# Belief state representation in the dopamine system

Benedicte M. Babayan[1,2], Naoshige Uchida [1] & Samuel.J. Gershman [2]

Learning to predict future outcomes is critical for driving appropriate behaviors. Reinforcement learning (RL) models have successfully accounted for such learning, relying on reward prediction errors (RPEs) signaled by midbrain dopamine neurons. It has been proposed that when sensory data provide only ambiguous information about which state an animal is in, it can predict reward based on a set of probabilities assigned to hypothetical states (called the belief state). Here we examine how dopamine RPEs and subsequent learning are regulated under state uncertainty. Mice are first trained in a task with two potential states defined by different reward amounts. During testing, intermediate-sized rewards are given in rare trials. Dopamine activity is a non-monotonic function of reward size, consistent with RL models operating on belief states. Furthermore, the magnitude of dopamine responses quantitatively predicts changes in behavior. These results establish the critical role of state inference in RL.

[1] Department of Molecular and Cellular Biology, Center for Brain Science, Harvard University, 16 Divinity Avenue, Cambridge, MA 02138, USA. [2] Department of Psychology, Center for Brain Science, Harvard University, 52 Oxford Street, Cambridge, MA 02138, USA. These authors contributed equally: Naoshige Uchida, Samuel J. Gershman. Correspondence and requests for materials should be addressed to N.U. (email: uchida@mcb.harvard.edu) or to Samuel.J.G. (email: gershman@fas.harvard.edu)

D opamine neurons are thought to report a reward prediction error (RPE, or the discrepancy between observed and predicted reward) that drives updating of predictions[1–5]. In reinforcement learning (RL) theories, future reward is predicted based on the current state of the environment[6]. Although many studies have assumed that the animal has a perfect knowledge about the current state, in many situations the information needed to determine what state the animal occupies is not directly available. For example, the value of foraging in a patch depends on ambiguous sensory information about the quality of the patch, its distance, the presence of predators, and other factors that collectively constitute the environment's state.

Normative theories propose that animals represent their state uncertainty as a probability distribution or belief state[7–10] providing a probabilistic estimate of the true state of the environment based on the current sensory information. Specifically, optimal state inference as stipulated by Bayes' rule computes a probability distribution over states (the belief state) conditional on the available sensory information. Such probabilistic beliefs about the current's state identity can be used to compute reward predictions by averaging the state-specific reward predictions weighted by the corresponding probabilities. Similarly to the way RL algorithms update values of observable states using reward prediction errors, state-specific predictions of ambiguous states can also be updated by distributing the prediction error across states in proportion to their probability. Simply put, standard RL algorithms compute reward prediction on observable states, but under state uncertainty reward predictions should normatively be computed on belief states, which correspond to the probability of being in a given state.

This leads to the hypothesis that dopamine activity should reflect prediction errors computed on belief states. However, direct evidence for this hypothesis remains elusive. Here we examine how dopamine RPEs and subsequent learning are regulated under state uncertainty, and find that both are consistent with RL models operating on belief states.

## Results

**Testing prediction error modulation by belief state**. We designed a task that allowed us to test distinct theoretical hypotheses about dopamine responses with or without state inference. We trained 11 mice on a Pavlovian conditioning task with two states distinguished only by their rewards: an identical odor cue predicted the delivery of either a small ($s_1$) or a big ($s_2$) reward (10% sucrose water) (Fig. 1a). The different trial types were presented in randomly alternating blocks of five identical trials, and a tone indicated block start. Only one odor and one sound cue was used for all blocks, making the two states perceptually similar prior to reward delivery. This task feature resulted in ambiguous sound and odor cues, since they were themselves insufficiently informative of the block identity, rendering the two states ambiguous with respect to their identity. This feature increased the likelihood of mice relying on probabilistic state inference.

To test for state inference influence on dopaminergic neuron signaling, we then introduced rare blocks with intermediate-sized rewards. Because the same odor preceded both reward sizes, a standard RL model with a single state would produce RPEs that increase linearly with reward magnitude (Fig. 1b, Supplementary Fig. 1a)[11, 12]. This prediction follows from the fact that the single state's value will reflect the average reward across blocks, and RPEs are equal to the observed reward relative to this average reward value. The actual value of the state will affect the intercept of the linear RPE response, but not its monotonicity. In Fig. 1b and Supplementary Fig 1a, we illustrated our prediction with a

state $s_t$ of average value 0.5 (on a scale between 0 and 1, which would be equivalent to 4.5 μL).

A strikingly different pattern is predicted by an RL model that uses state inference to compute reward expectations. Optimal state inference is stipulated by Bayes' rule, which computes a probability distribution over states (the belief state) conditional on the available sensory information. This model explicitly assumes the existence of multiple states distinguished by their reward distributions (see methods). Thus, in spite of identical sensory inputs, prior experience allows to probabilistically distinguish several states (one associated to 1 μL and one to 10 μL). If mice rely on a multi-state representation, they now have two reference points to compare the intermediate rewards to. Upon the introduction of new intermediate rewards, the probability of being in the state $s_1$ would be high for small water amounts and low for large water amounts (Fig. 1c). The subsequent reward expectation would then be a probability-weighted combination of the expectations for $s_1$ and $s_2$. Consequently, smaller intermediate rewards would be better than the expected small reward (a positive prediction error) and bigger intermediate rewards would be worse than the expected big reward (a negative prediction error), resulting in a non-monotonic pattern of RPEs across intermediate rewards (Fig. 1d, Supplementary Fig. 1c).

In our paradigm, because reward amount defines states, reward prediction and belief state are closely related. Yet with the same reward amount, standard RL and belief state RL make qualitatively different predictions (Fig. 1b, d). The main distinction between both classes of models is the following: the standard RL model does not have distinct states corresponding to the small and large reward states, and reward prediction is based on the cached value learned directly from experienced reward, whereas the belief state model has distinct states corresponding to the small and large reward states (Supplementary Fig. 1, left column). In the latter case, the animal or agent uses ambiguous information to infer which state it is in, and predicts reward based on this inferred state (i.e., belief state).

To test whether dopamine neurons in mice exhibited this modulation by inferred states, we recorded dopamine neuron population activity using fiber photometry (fluorometry) (Fig. 1e)[13–16]. We used the genetically encoded calcium indicator, GCaMP6f[17, 18], expressed in the ventral tegmental area (VTA) of transgenic mice expressing Cre recombinase under the control of the dopamine transporter gene (DAT-cre mice)[19] crossed with reporter mice expressing red fluorescent protein (tdTomato) (Jackson Lab). We focused our analysis on the phasic responses. Indeed, calcium imaging limits our ability to monitor long-timescale changes in baseline due to technical limitations such as bleaching of the calcium indicator. Moreover a majority of previous work studying dopamine neurons has shown reward prediction error-like signaling in the phasic responses[1, 3, 12]. Similarly to single-cell recordings[1, 3, 12], population activity of dopamine neurons measured by fiber photometry in the VTA[20] (Supplementary Fig. 2) or in terminals of dopamine neurons projecting to the ventral striatum[16, 21] show canonical RPE coding in classical conditioning tasks.

**Behavior and dopamine neuron activity on training blocks**. After training mice on the small ($s_1 = 1$ μL) and big ($s_2 = 10$ μL) states, we measured their amount of anticipatory licking, a readout for reward expectation, and the dopamine responses (Fig. 2a, d). At block transitions, mice had a tendency to anticipate a change in contingency as they increased anticipatory licking in trial 1 following a small block (one sample $t$-tests, $p < 0.05$, Fig. 2b), leading to similar levels of anticipatory licking on trial 1

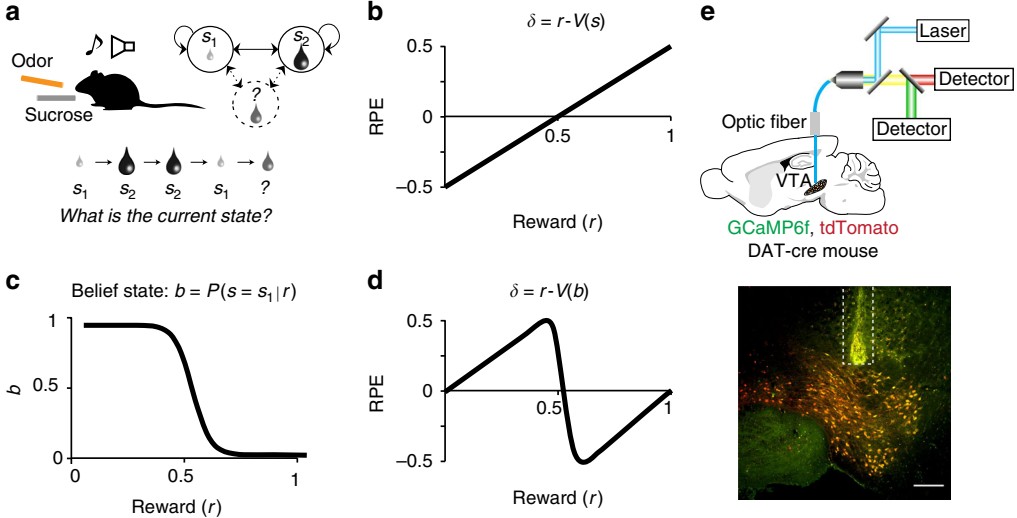

**Fig. 1** Task design to test the modulation of dopaminergic RPEs by state inference. **a** Mice are trained on two perceptually similar states only distinguished by their rewards: small ($s_1$) or big ($s_2$). The different trial types, each starting by the onset of a unique odor (conditioned stimulus, CS) predicting the delivery of sucrose (unconditioned stimulus, US), were presented in randomly alternating blocks of five identical trials. A tone indicated block start. Only one odor and one sound cue were used for all blocks, making the two states perceptually similar prior to reward delivery. To test for state inference influence on dopaminergic neuron signaling, we then introduced rare blocks with intermediate-sized rewards. Using (with reinforcement learning (RL) operating on belief states) or not (with standard RL) the training blocks as reference state for computing the value of the novel intermediate states predicts contrasting RPE patterns (**b** vs **d**). **b** RPE across varying rewards computed using standard RL. Because the same odor preceded both reward sizes, a standard RL model with a single state would produce RPEs that increase linearly with reward magnitude. **c** Belief state $b$ across varying rewards defined as the probability of being in $s_1$ given the received reward. **d** RPE across varying rewards computed using the value of the belief state $b$. A non-monotonic pattern across increasing rewards is predicted when computing the prediction error on the belief state $b$. **e** (Top) Population activity of VTA dopaminergic neurons is recorded in behaving mice using fiber photometry. (Bottom) Fiber location above the recorded cells in the VTA, which co-express the calcium reporter GCaMP6f and the fluorescent protein tdTomato (scale bar: 200 μm)

(two-way ANOVA, no effect of current or previous block, $p > 0.16$; Fig. 2a, c). The dopamine response on cue presentation did not show such modulation, only reflecting the activity on the previous trial (one sample $t$-tests, $p > 0.27$, Fig. 2e; two-way ANOVA, main effect of previous block on trial 1, $p = 0.0025$, Fig. 2f), although the response on reward presentation showed modulation by both the current and previous block (two-way ANOVA on trial 1, main effect of current block, $p < 0.001$, main effect of previous block, $p = 0.038$, Fig. 2h), with significant changes in amplitude at block transitions for block $s_1$ following $s_2$ and blocks $s_2$ (one sample $t$-tests, $p < 0.01$, Fig. 2g).

Analyzing the licking and dopamine activity at block start, when the sound comes on, mice appeared to increase licking following the small block $s_1$ between sound offset and trial 1's odor onset (during a fixed period of 3 s) (Supplementary Fig. 3a, b). Although this was not sufficient to actually reverse the licking pattern on trial start, it likely contributed to the observed change in licking between trial 5 and 1 (Fig. 2b). Dopamine activity showed the opposite tendency, with decreasing activity following blocks $s_2$ (Supplementary Fig. 3c, d). This activity on block start indicated that mice partially predicted a change in contingency, following the task's initial training structure (deterministic switch between blocks during the first 10 days). However, this predictive activity did not override the effect of the previous block on dopamine activity on cue presentation as it was most similar to the activity on the preceding block's last trial (Fig. 2e). Following trial 1, anticipatory licking and dopamine activity on cue and reward presentation reached stable levels, with lower activity in $s_1$ compared to $s_2$ (two-way ANOVAs, main effect of current block on trials 2 to 5, $p < 0.05$, no effect of previous block, $p > 0.4$, nor interaction, $p > 0.5$; Fig. 2c, f, h). The stability in anticipatory licking and dopamine activity after exposure to the first trial of a block

suggested that mice acquired the main features of the task: reward on trial 1 indicates the current block type and reward is stable within a block.

**Dopaminergic and behavioral signature of belief states.** Once mice showed a stable pattern of licking and dopamine neuron activity in the training states (Fig. 2), every other training day we replaced 10% of the training blocks (3) by intermediate reward blocks, with each intermediate reward being presented no more than once per day. Over their whole training history, each mouse experienced $3980 \pm 213$ (mean ± s.e.m.) trials of each training block and $42 \pm 6$ (mean ± s.e.m.) trials of each intermediate reward (Supplementary Fig. 4). On the first trial of reward presentation, the dopamine neurons responded proportionally to reward magnitude (Fig. 3a–c). Importantly, the monotonically increasing response on this first trial, which informed mice about the volume of the current block, suggested dopamine neurons had access to the current reward. On the second trial, the response of dopamine neurons presented a non-monotonic pattern, with smaller responses to intermediate rewards (2 and 4 μL) than to bigger intermediate rewards (6 and 8 μL) (Fig. 3e, f, g).

These monotonic and non-monotonic patterns on trials 1 and 2, respectively, were observed in our three different recording conditions: (1) in mice expressing GCaMP6f transgenetically in DAT-positive neurons and recorded from VTA cell bodies ($n = 5$), (2) in mice expressing GCaMP6f through a viral construct in DAT-positive neurons and recorded from VTA cell bodies ($n = 2$); (3) in mice expressing GCaMP6f through a viral construct in DAT-positive neurons and recorded from dopamine neuron terminals in the ventral striatum ($n = 4$) (Supplementary Fig. 5a–c). Although these patterns were observed in each condition, the amplitude of the signal varied across the different recording conditions, largely due to lower expression levels of

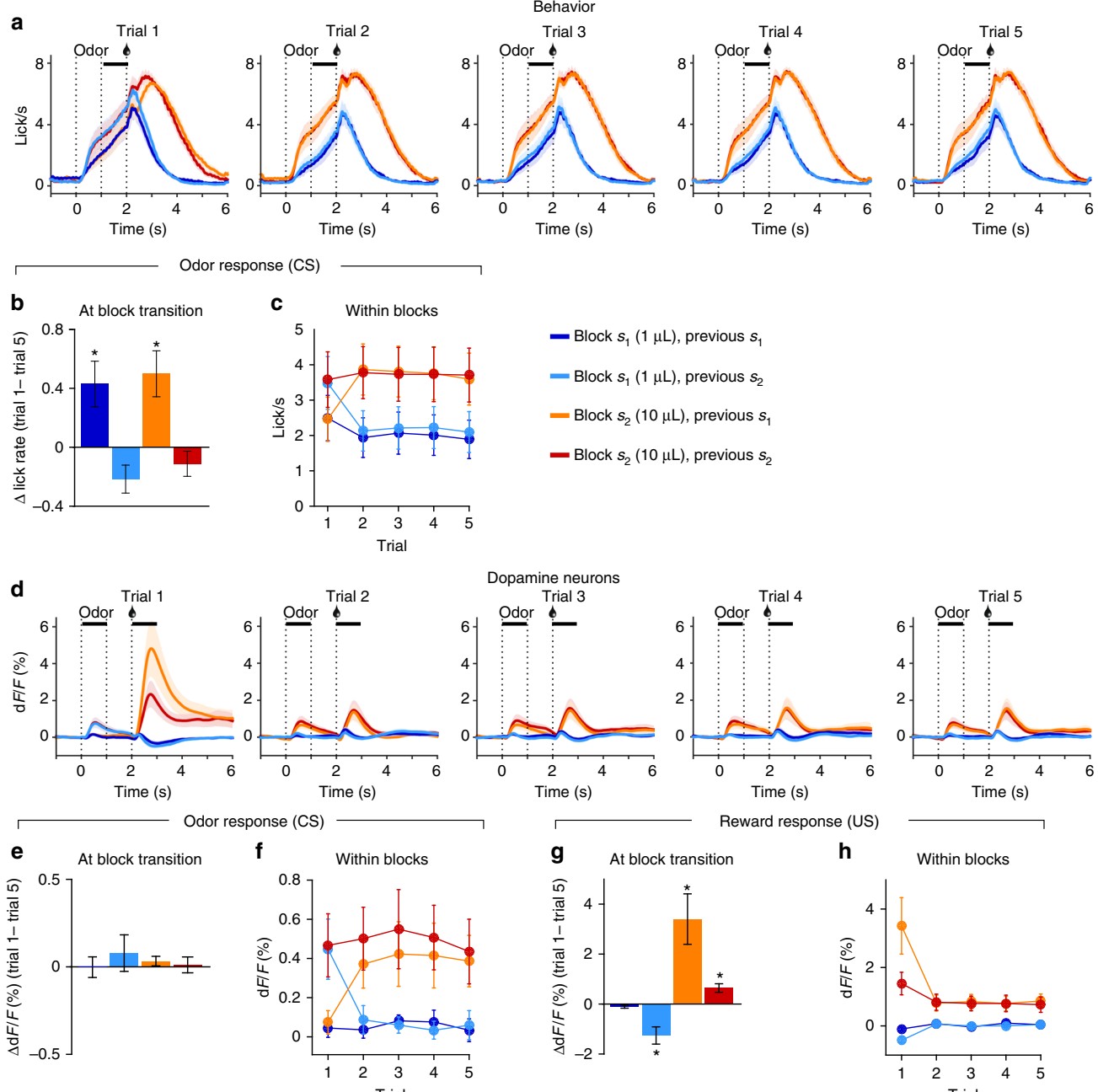

**Fig. 2** Behavior and dopamine neuron activity on training blocks $s_1$ and $s_2$. **a** Licking across the five trials within a block. Anticipatory licking quantification period during odor to reward delay is indicated by the horizontal black line. **b** Anticipatory licking at block transition increases when transitioning from the small to the big block. **c** Anticipatory licking across trials within blocks. Anticipatory licking on trial 1 is similar across all block types then stabilizes at either low or high rates for the following four trials. **d** Dopamine neuron activity across the five trials within a block. Horizontal black line indicates quantification period for odor (CS) and reward (US) responses. **e** Dopamine neurons odor response across block transitions is stable. **f** Dopamine neurons odor response across trials. Dopamine activity adapts to the current block value within one trial. **g** Dopamine neurons reward response shows an effect of the current reward and previous block on trial 1. **h** Dopamine neurons reward response across trials. Dopamine activity reaches stable levels as from trial 2. Data represents mean ± s.e.m. *$p < 0.05$ for $t$-test comparing average value to 0. $n = 11$ mice

GCaMP in transgenic mice compared to those with viral expression and overall variability in signal intensity across animals within each recording condition. Therefore, for illustration purposes, we normalized the signals from each individual mouse using trial 1's response as reference for the minimum and maximum values for the min–max normalization ($y = (x - \min_{\text{trial1}})/(\max_{\text{trial1}} - \min_{\text{trial1}})$) to rescale the GCaMP signals in the 0 to 1 range (Supplementary Fig. 5d–f, Figs. 3 and 4). Similar

results were obtained when measuring the peak response following reward presentation instead of the average activity over 1 s (Supplementary Fig. 6a–g).

We compared the fits of linear and polynomial functions to the dopamine responses, revealing highest adjusted $r^2$ for a linear fit for trial 1 (Supplementary Fig. 7a) and for a cubic polynomial fit for trial 2 (Supplementary Fig. 7b). The non-monotonic pattern observed on trial 2 was consistent with our hypothesis of belief

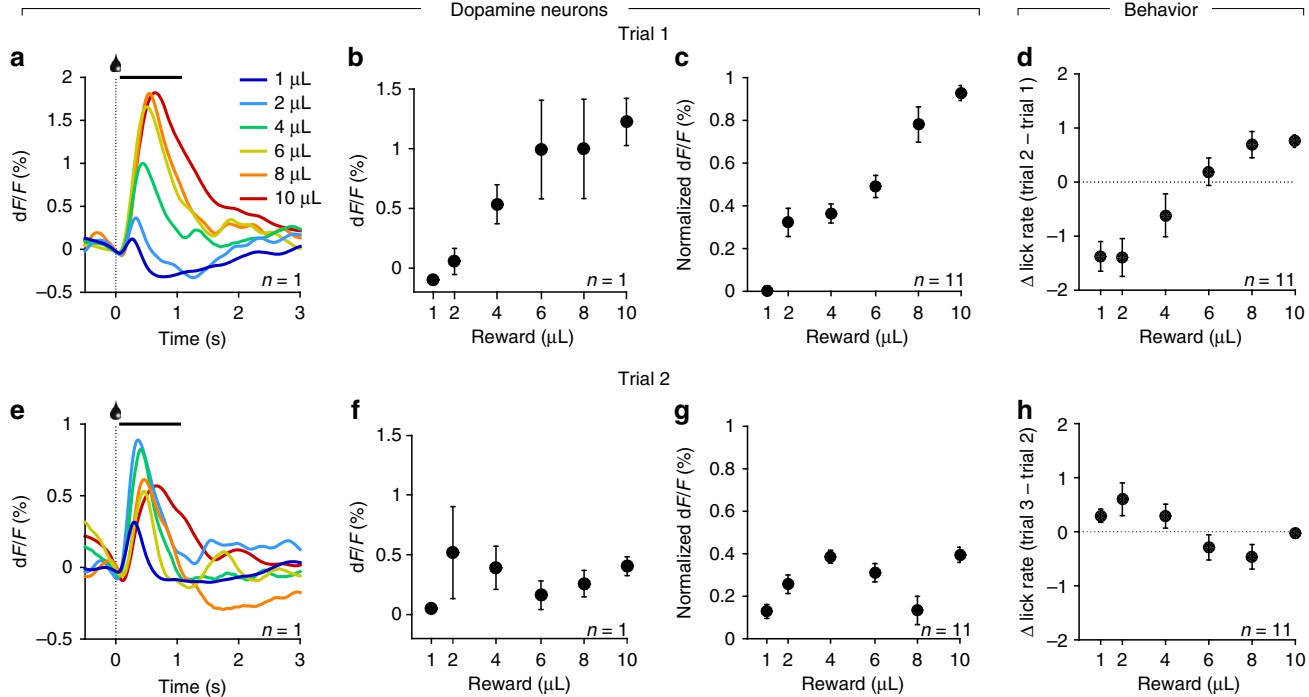

**Fig. 3** Dopaminergic and behavioral signature of belief states. **a–c** Dopamine neurons activity on trial 1. Dopamine neurons show a monotonically increasing response to increasing rewards (**a**, individual example), quantified as the mean response after reward presentation (0–1 s, indicated by a solid black line in **a**) in the individual example (**b**) and across mice (**c**). **d** Change in anticipatory licking from trial 1 to trial 2. Mice increase their anticipatory licking after trial 1 proportionally to the increasing rewards. **e–g** Dopamine neurons activity on trial 2. Dopamine neurons show a non-monotonic response pattern to increasing rewards (**e**, **f**, individual example), quantified across all mice (**g**). **h** Change in anticipatory licking from trial 2 to trial 3. Whereas mice do not additionally adapt their licking for the known trained volumes (1 and 10 μL) after trial 2, they increase anticipatory licking for small intermediate rewards and decrease it for larger intermediate rewards in a pattern, which follows our prediction of belief state influence on RPE. $n = 11$, data represent mean ± s.e.m.

state influence on dopamine reward RPE (Fig. 1d). We focused our analysis on trial 2 since, according to our model, that is the most likely trial to show an effect of state inference with the strongest difference from standard RL reward prediction errors (Supplementary Fig. 8a, b). Both RL models predict weaker prediction error modulation with increasing exposure to the same reward and we observed weaker versions of this non-monotonic pattern in later trials (Supplementary Fig. 8a, c). It is however interesting to note that different mice showed a non-monotonic reward response modulation at varying degrees on distinct trials. For example, Mouse 4 showed a strong non-monotonic pattern on trial 2, which then became shallower on the following trials, whereas Mouse 9 showed a more sustained non-monotonic pattern across trials 2 to 5 (Supplementary Fig. 8d). Lastly, the pattern of dopamine responses was observed independently of the baseline correction method we used, whether it was pre-trial, pre-block, or using a running median as baseline (Supplementary Fig. 9).

We next analyzed whether behavior was influenced by state inference. Anticipatory licking before reward delivery is a read-out of mice's reward expectation. Dopamine RPEs are proposed to update expectations. To test whether mice's behavioral adaptation across trials followed the dopaminergic RPE pattern, we measured how mice changed their anticipatory licking across trials. From trial 1 to trial 2, mice changed their anticipatory licking proportionally to the volume (Fig. 3d) but showed a non-monotonic change from trial 2 to trial 3 (Fig. 3h; highest adjusted $r^2$ for a cubic polynomial fit, Supplementary Fig. 7d). Fits of linear and polynomial functions to the change in anticipatory licking revealed highest adjusted $r^2$ for cubic polynomial fits for both transitions from trial 1 and 2 (Supplementary Fig. 7c), although

the linear fit still provided a decent fit (adjusted $r^2 = 0.94$). Thus, dopamine activity and change in anticipatory licking both showed modulation according to our prediction of the influence of belief state on RPE (Fig. 1d). Although the average change in anticipatory licking for transitions from trial 3 to 5 did not seem to visibly follow the pattern of dopamine activity (Supplementary Fig. 10a), a trial-by-trial analysis showed that dopamine responses on reward presentation were significantly correlated with a change of licking on following trial for all trial transitions within blocks (trial 1 to 5, Pearson's $r$, $p < 2.5 \times 10^{-3}$, Supplementary Fig. 10b), suggesting that inhibition or lower activations of dopamine neurons were more often followed by a decrease in anticipatory licking whereas transient activations of dopamine neurons tended to be followed by increased anticipatory licking.

**Belief state RL explains dopamine responses and behavior.** We next tested whether an RL model operating on belief states could explain the dopamine recordings better than a standard RL model. As the odor indicating trial start was identical for all reward sizes, a standard RL model (without belief states) would assume a single state, with prediction errors that scale linearly with reward (Supplementary Fig. 1a). An RL model using belief states, by contrast, differentiates the states based on the current reward size and the history of prior reward sizes within a block (Supplementary Fig. 1c). Belief states were defined as the posterior probability distribution over states given the reward history, computed using Bayes' rule (Methods). Since the previous block had an effect on the expectation of the first trial of a given block (Fig. 2), we allowed for two different initial values on block start depending on the previous block in both models, and fit RL models to the trial-by-trial dopamine response of each trained

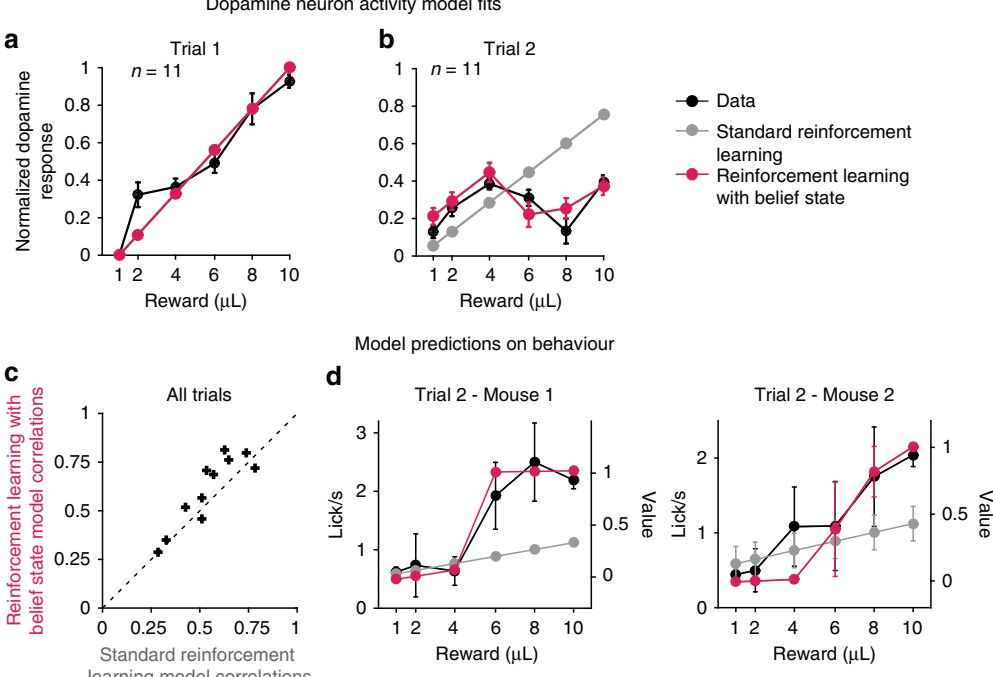

**Fig. 4** RL with belief states explains dopamine reward responses and behavior better than standard RL. Individual DA responses to rewards were fit using either a standard RL model or a RL model computing values on belief states. **a** Fits to dopamine responses on trial 1. Both RL models fit the dopamine response, since on trial 1 there is no evidence to infer a state on. **b** Fits to dopamine responses on trial 2. Only computing RPEs using belief states reproduced the non-monotonic change in dopamine response across increasing rewards. **c** Model predictions on behavior. The value functions from either model fits were positively correlated with the mice's anticipatory licking, but the RL model with belief state provided a better fit (signed rank test: $p = 0.032$), suggesting that mice's anticipatory licking tracks the value of the belief state. **d** Individual examples of extracted value function from either model and anticipatory licking across increasing rewards on trial 2. $n = 11$, data represent mean ± s.e.m.

mouse across all trials (Supplementary Fig. 1b, e). On trial 1, both models predicted and fit the linearly increasing dopamine response to increasing rewards (Fig. 4a). On trial 2, only RPEs computed on belief states reproduced the non-monotonic change in dopamine response across increasing rewards (Fig. 4b). We additionally tested four model variants, two that did not include influence from the previous block on the value for the standard RL model (Supplementary Fig. 1a) or on the prior at block start for the belief state model (Supplementary Fig. 1c), as well as two other variants of the belief state RL model with distinct priors based on the previous block (Supplementary Fig. 1d) or with three states, adding a belief state for the intermediate rewards (Supplementary Fig. 1f). Overall, only models computing prediction errors on belief states could qualitatively reproduce the non-monotonic pattern of dopamine activity on trial 2 (Supplementary Fig. 1g–l). Bayesian information criterion (BIC) and random-effects model selection[22, 23] computed on each of the six models fit to individual mice's dopamine activity both favored the RL model with belief states with two initial free priors over other models, in particular over the standard RL model with two free initial values (Supplementary Table 1; Supplementary Fig. 8c). Similar results were obtained when fitting the peak GCaMP response after reward presentation (Supplementary Table 2; Supplementary Fig. 6h).

Since anticipatory licking in the training blocks reflected the value of each training block (Fig. 2c), we next examined the relationship between anticipatory licking and values in the RL models, with or without belief states, which obtained the best model comparison scores (BIC and protected exceedance probability). The models were not fit to these data and hence this constitutes an independent test of the model predictions. For each mouse, anticipatory licking in all trials and all reward sizes

was positively correlated with values extracted from both RL models (one-tailed $t$-test, $p < 1.0 \times 10^{-6}$), but the correlations were significantly higher with the values computed using a RL model with belief state (Fig. 4c; Signed rank test, signed rank = 9, $p = 0.032$), and as shown in two individual examples (Fig. 4d). Although we only fit the model RPEs to the dopamine reward response, the belief state values used to compute the error term were apparent in the anticipatory licking activity. Finally, we performed the same analysis on the dopamine response at cue onset (Supplementary Fig. 11). Dopamine activity at cue onset appeared to follow a step function on trials 2 to 5 across increasing rewards (Supplementary Fig. 11a), similar to the predicted belief state value (Supplementary Fig. 1c–f). This activity was positively correlated with values from both models (one-tailed $t$-test, $p \leq 1.0 \times 10^{-3}$, Supplementary Fig. 11b), although no model was a significantly better predictor (Signed rank test, signed rank = 21, $p = 0.32$).

## Discussion

Our results suggest that mice make inferences about hidden states based on ambiguous sensory information, and use these inferences to determine their reward expectations. In our task design, this results in a non-monotonic relationship between reward magnitude and RPE, reflected in the response of dopamine neurons. Although this pattern is strikingly different from the patterns observed in classical conditioning studies[12, 24, 25], it can be qualitatively and quantitatively accommodated by a model in which RPEs are computed on belief states. Our results complement recent studies that have provided additional evidence for reflections of hidden-state inference in dopamine responses, for example when animals learn from ambiguous temporal[26–28] and visual[29] cues.

Two features of our task design allowed us to specifically test the influence of belief states on dopamine RPE: an extended training on two reference states, which allowed mice to build a strong prior over reward distributions, and ambiguity in the cues used to signal upcoming reward combined to inherent uncertainty in the sensory perception of water amount. Importantly, the intensive training on the two reference states did not alter the ability of dopamine neurons to discern the new intermediate reward sizes when first exposed to them, so the observed non-monotonic pattern is unlikely to be explained by biased sensory processing. Interestingly, both anticipatory licking and dopamine activity appeared to predict a switch in contingency upon block start. Although the amplitude of these pre-emptive changes were relatively small compared to responses to the odor cue and reward presentations, it indicated that the task structure influenced both behavior and dopamine activity, as had been previously shown in macaques[30].

Increasing evidence suggests that dopamine neurons that project to the dorsal striatum signal different types of signals. Indeed dopamine neurons projecting to specific regions of the dorsal striatum have been shown to be activated by rewarding, aversive and novel stimuli[16, 31, 32]. Here we recorded from the canonical dopamine system, involving VTA to ventral striatum loops, which encode value prediction errors[16, 33, 34]. Whether other dopamine inputs projecting to other areas of the dorsal striatum and broadcasting different types of signals can also be modulated by inferred states remains to be addressed.

The exact sources of calcium signals remain unclear. Most, if not all, of in vivo calcium imaging studies assume that large calcium influxes through voltage-gated calcium channels evoked by spikes dominate calcium signals that they measure. Nonetheless, this might not be true in some systems. With respect to the dopamine system, there are some unique points that need to be taken into account when we interpret calcium imaging data. First, dopamine neurons have a mechanism to maintain the baseline, pace-making activity, which relies on calcium[35]. Second, increasing evidence suggests that dopamine release is regulated at the level of axon terminals, through cholinergic and glutamatergic mechanisms[36–38]. Furthermore, cholinergic interneurons in the dorsomedial striatum have been shown to track beliefs about current state[39]. However, because our main results hold whether we monitored the activity from cell bodies or axons of dopamine neurons, these additional processes are unlikely to affect our observation of state inference modulation of dopamine neuron activity.

An important question for future research is to determine the origins of belief state inputs into the dopamine system. One potential substrate is the orbitofrontal cortex, which has been proposed to encode state spaces, in particular when states are perceptually similar but conceptually different[40]. Dopamine RPEs have also been shown to be influenced by inferred states in reversal[30] and sensory-preconditioning tasks[41], which appear to rely on state inference and require an intact orbitofrontal cortex[42–45]. Another potential substrate for belief state inference is the hippocampus. It has been proposed to support structure learning[46–50], which would allow mice to infer the latent causes governing the structure of a task, such as learning the two-state representation despite ambiguous predictive cues. A recent study found that dopamine neurons alter their responses based on changes in sensory features of reward[51]. In the present study, we focused on reward prediction errors based on reward sizes. It would be interesting to extend the present study using different sensory features (e.g., taste or smell of reward) that may define "states" in multiple dimensions, which may in turn recruit distinct partners for computing beliefs regarding their identity.

In summary, our data provide direct support for the hypothesis that belief states can drive dopamine RPEs, and subsequent behavioral learning when animals are uncertain about the current state. Although RL accounts of dopamine have typically conceptualized its computational function as "model-free"[52], our data suggest that an internal model of the environment may have a central role in governing dopamine responses.

## Methods

**Animals.** Eleven adult male mice were used. All mice were heterozygous for Cre recombinase under the control of the *DAT* gene (B6.SJL-Slc6a3tm1.1(cre)Bkmn/J; Jackson Laboratory)[19], crossed to *Rosa26-tdTomato* reporter mice (Ai9, JAX 007909). Five mice were crossed to Ai95D (*Rosa26-GCaMP6f* reporter mice, JAX 024105). All mice were housed on a 12 h dark/12 h light cycle (dark from 06:00–18:00) and each performed the behavioral task at approximately the same time of day each day. After surgery they were individually housed. All surgical and experimental procedures were in accordance with the National Institutes of Health Guide for the Care and Use of Laboratory Animals and approved by the Harvard Institutional Animal Care and Use Committee.

**Surgery.** All surgeries were performed under aseptic conditions with animals under isoflurane (1%–2% at 1 L/min) anesthesia. Analgesia (ketoprofen, 5 mg/kg, I.P.; buprenorphine, 0.1 mg/kg, I.P.) was administered preoperatively and post-operatively for 48 h. Mice were surgically implanted with a custom head-plate[3] and an optic fiber either above the medial VTA to record from cell bodies (Bregma −3.1 AP, 0.6 ML, 4.3 DV; n = 7) or in the ventral striatum to record from dopamine neurons terminals (Bregma 1.6 AP, 1.3 ML, 3.75 DV; n = 4). No difference was observed in the signal obtained from either region. The head-plate was affixed to the skull with dental cement (C&B Metabond) and the optic fiber (200 µm diameter, Doric Lenses) was secured using UV-curing epoxy (Thorlabs, NOA81), followed by a layer of black Ortho-Jet dental adhesive (Lang Dental). During the same surgery, the 6 mice not crossed with GCaMP6f reporter mice received 200–400 nL of AAV9/Syn-Flex-GCaMP6f (Upenn Vector Core, diluted 4× in HBSS) injections into the VTA (Bregma −3.1 AP, 0.6 ML, 4.3 DV).

**Behavioral paradigm.** After 1 week of recovery, mice were water-restricted in their cages. Weight was maintained above 85% of baseline body weight. Animals were head-restrained and habituated for 1–2 days before training. Odors were delivered with a custom-made olfactometer[53]. Each odor was dissolved in mineral oil at 1:10 dilution. 30 µL of diluted odor was placed inside a filter-paper housing, and then further diluted with filtered air by 1:20 to produce a 1000 mL/min total flow rate. Odors included isoamyl acetate, 1-hexanol and caproic acid, and differed for different animals. Licks were detected by breaks of an infrared beam placed in front of the water tube (n.b. the licking behavior had no effect on whether water was delivered).

Trials were presented in blocks of 5 trials. A 15 kHz tone lasting 2 s signaled block start, ending 3 s before the start of a block's first trial. Each trial began with 1 s odor delivery (one odor per mouse), followed by a 1 s delay and an outcome (1 to 10 µL of 10% sucrose water, constant within a block). Inter-trial intervals were on average 8.7 s, composed of an initial fixed 4 s period, to ensure GCaMP signals went down to baseline between trials, followed by an interval drawn from an exponential distribution (mean: 4.7 s), resulting in a flat hazard function such that mice had constant expectation of when the next trial would begin. Mice did 30 blocks per day (150 trials).

Mice were trained 10 to 15 days on a deterministic training regime, with alternating small ($s_1$, 1 µL) and big ($s_2$, 10 µL) blocks. The transition between blocks then became probabilistic, with a 50% probability of block change when a block started. Intermediate reward blocks (2, 4, 6, and 8 µl) were introduced only after >20 days of training, every other training day. When mice were probed on intermediate rewards, 3 (10%) of the training blocks were swapped by 3 different intermediate reward block. The 11 mice were trained on this task. There are no distinct experimental groups in this study, so no randomization or blinding was required.

For the classical conditioning task (Supplementary Fig. 2), one mouse was trained to associate 3 different odors to three reward probabilities (0%, 50%, 90%). The trials were presented pseudo-randomly, interspersed with 10% of unpredicted water delivery, performing 200 to 300 trials per day.

**Fiber photometry.** The fiber photometry (or fluorometry) system used blue light from a 473 nm DPSS laser (80–500 µW; Opto Engine LLC, UT, USA) filtered through a neutral density filter, Thorlabs, NJ, USA) and coupled into an optical fiber patchcord (400 µm, Doric Lenses, Quebec, Canada) using a 0.65 NA microscope objective (Olympus). The patchcord connected to the implanted fiber simultaneously delivered excitation light and collected fluorescence emission. Activity-dependent fluorescence emitted by cells in the vicinity of the implanted fiber tip was spectrally separated from the excitation light using a dichroic mirror (Chroma, NY, USA), passed through a band pass filter (ET500/50,

Chroma) and focused onto a photodetector (FDS10X10, Thorlabs) connected to a current preamplifier (SR570, Stanford Research Systems). Acquisition from the red (tdTomato) fluorophore was simultaneously acquired (band pass filter ET605/70 nm, Chroma). The preamplifier output voltage signal was collected by a NIDAQ board (PCI-e6321, National Instruments) connected to a computer running Lab-VIEW (National Instruments) for signal acquisition.

We have examined whether our signals contain motion artefacts in a previous study[16]. Using a set-up with 473 and 561 nm lasers to deliver light to excite respectively GFP and tdTomato reporters, we previously observed large responses to unpredicted reward in GCaMP, but not tdTomato, signals when mice are head-fixed. We thus did not correct the GCaMP signals with tdTomato signals.

**Anatomical verification**. At the end of training, mice were given an overdose of ketamine/medetomidine, exsanguinated with saline, perfused with 4% paraf-ormaldehyde, and brains were cut in 50 or 100 µm coronal sections. Sections were stained with 4′,6-diamidino-2-phenylindole (DAPI) to visualize nuclei. Recording sites and GCaMP6f expression were verified to be amid tdTomato expression in dopamine neurons cell bodies or ventral striatum terminals (Fig. 1e).

**Data analysis**. Lick rate was acquired at 1 kHz. Mean anticipatory licking was calculated for each trial as the mean lick rate in the 1 s delay period between odor presentation and water delivery. The differential lick rate (Δ lick rate) was computed as the difference of mean anticipatory licking between two consecutive trials, within a training day.

For GCaMP activity, we focused our analysis on the phasic responses. Indeed, a majority of previous work has shown reward prediction error-like signaling in the phasic responses of dopamine neurons and technical limitations such as bleaching limit our ability to monitor long-timescale changes in baseline using calcium imaging. Fluorescence data was acquired at 1 kHz. For each trial, the relative change in fluorescence, $dF/F = (F - F0)/F0$, was calculated by taking $F0$ to be the mean fluorescence during a 1 s period before the odor presentation, such that the fluorescence measured at each time point within a trial is corrected by the average fluorescence during the 1 s period before odor presentation for that given trial. We further tested two additional baseline normalizations to verify that our conclusions were robust with regards to the baseline normalization method (Supplementary Fig. 9): (1) using as F0 the 1 s period before block start, i.e., before sound onset, such that the fluorescence measured at each time point within a trial is corrected by the average fluorescence during the 1 s period before sound presentation for that given block (i.e., over five consecutive trials); (2) using as F0 the median over a 60 s window, such that the fluorescence measured at each time point is corrected by the median fluorescence over a 60 s period centered around that given time point.

Mean GCaMP activity during odor (CS) and reward (US) presentations was calculated for each trial as the mean activity during the 1 s period after event onset. Data and model fitting were additionally verified with the peak GCaMP activity following the reward response, by quantifying the maximum response in the 1 s window after reward delivery (Supplementary Fig. 6, Supplementary Table 2). Two types of further normalization were performed on the data, regardless of the baseline correction used: (1) When analyzing the reward (US) response, since the CS response did not always go back to baseline before reward presentation, US responses were baseline-corrected by subtracting the mean $dF/F$ over the 100 ms period centered around US onset. This provided a measure for the actual change in activity at reward presentation. (2) Since the absolute level of fluorescence was variable across mice that expressed GCaMP6f through viral injection or transgenetically (Supplementary Fig. 5), for illustration purposes to summarize the data in one plot, each mouse's mean US response across rewards was normalized by min–max normalization when pooled together. The normalization was performed within each mouse, using the given mouse's trial 1 response as reference for the minimum and maximum values for the min–max normalization such that $y = (x - \mathrm{min}_{\mathrm{trial1}})/(\mathrm{max}_{\mathrm{trial1}} - \mathrm{min}_{\mathrm{trial1}}))$ (Fig. 3c and g, Supplementary Figs. 5–11). Of note, the models were not fit on the min–max normalized data but directly on mice's individual baseline-corrected GCaMP activity.

Polynomial fits to the dopamine neuron activity and behavior were performed using the polyfit function in MATLAB.

**Computational modeling**. Standard RL: We used a simplified version of the temporal difference (TD) model[11], modeling stimuli and rewards at the trial level instead of in real time. This model learned values ($V$) for each state ($s$). In our task, states correspond to blocks ($s_1$ = small reward block, $s_2$ = large reward block). The values were updated using the RPE

$$\delta_t = r_t - V(s_t),$$

following an observation of $r_t$, the reward delivered at trial t:

$$V(S_{t+1}) = V(S_t) + \alpha\delta_t,$$

where $\alpha$ is a learning rate and $r_t \in \{0, 1\}$, with $r_t = 0$ for the small reward block $s_1$ (1 µL) and $r_t = 1$ for the big reward block $s_2$ (10 µL).

The state $s$ was defined by the sensory input at trial start, the CS odor. Since the same odor preceded both reward sizes, a standard TD model would assume a single

state. We set that value at 0.5, the averaged reward over mice's reward history (Fig. 1b, Supplementary Fig. 1a). To account for the effect of the previous block on mice's expectations (Fig. 2), we also explored a version of this model where the value on trial 1 at block start could be different depending on the previous block ($s_1$ or $s_2$) (Fig. 4, Supplementary Fig. 1b).

RL with belief states: We used the same value learning rules as for the standard TD model but replaced the state by a belief state $b(s)$, which expresses the animal's state uncertainty as a probability distribution over states. This model assumed that on each trial, mice computed the posterior probability of being in a state $s$ given the observed reward $r$ following Bayes' rule:

$$b(s) = \frac{P(r|s)P(s)}{P(r)}.$$

The likelihood $P(r|s) = \mathrm{N}(r; \overline{r_s}, \sigma^2)$ was defined as a normal distribution over rewards $r$, centered on the average reward normally obtained in the current state $\overline{r_s}$ with a sensory noise variance $\sigma^2$ that captured uncertainty about the detected reward amount. This model thus explicitly assumed the existence of multiple states, distinguished only by their reward distributions. The prior $P(s)$ expressed the mice's prior about the likelihood of the occurrence of a given state. The denominator represented the marginal reward distribution across all states $P(r) = \sum_{s'} P(r|s')P(s')$.

Given the belief state $b$, the prediction error was:

$$\delta_t = r_t - V(b_t),$$

where the value function was approximated as a linear function of the belief state:

$$V(b_t) = w_1 b_t(s_1) + w_2 b_t(s_2).$$

Weights were then updated according to:

$$\Delta w = \alpha\delta_t b_t.$$

We tested four different versions of this model by testing different ways of setting the prior $P(s)$:

- Setting $P(s) = 0.5$ (Fig. 1c, d, Supplementary Fig. 1c), since the mice experienced $s_1$ and $s_2$ with equal probability during their training (Supplementary Fig. 4).
- Allowing $P(s)$ to be free parameters, defining $p_1 = P(s = s_1)$ as the prior following block $s_1$, and $p_2 = P(s = s_2) = 1 - p_1$ (Supplementary Fig. 1d).
- Allowing both $p_1$ and $p_2$ as free parameters (Fig. 4, Supplementary Fig. 1e).
- Setting 3 priors free: $p_1$, $p_2$ and an additional prior for intermediate state ($p_3$), which corresponded to mice building an additional state for the novel rewards (Supplementary Fig. 1f).

All belief state models had a minimum of three free parameters: the learning rate α, the sensory noise variance $\sigma^2$, and a coefficient $\beta$, which mapped theoretical prediction errors linearly to the measured dopamine response (i.e., the GCaMP signal). Indeed, because of a relatively long delay between odor onset and reward delivery (2 s), as well as timing jitter resulting from when mice first sniff after odor onset, we expected our dopamine reward responses to be generally shifted above 0[3, 4, 54]. This was accounted for by fitting $\beta$.

**Model fitting**. For each mouse, we computed the average dopamine response for each reward size and each trial (trials 1 to 5), separating the data based on the previous blocks. We fit the free parameters to the dopamine responses using maximum likelihood estimation. Optimization was performed using the MATLAB function fmincon, initializing the optimization routine at 5 random parameter values.

We used the following bounds on the parameter values:

- the learning rate $\alpha \in [0, 0.3]$,
- the sensory noise variance $\sigma^2 \in [0.01, 0.5]$,
- initial values $V \in [0, 1]$,
- priors $p \in [0.001, 0.999]$.

To compare model fits, we computed the Bayesian Information Criterion (BIC), which allows direct comparison between models that have different numbers of parameters, and exceedance and protected exceedance probabilities using Bayesian model selection analysis, which measure how likely it is that any given model is more frequent than all other models in the comparison set[22, 23].

**Code availability**. The models were programmed in MATLAB. The code is available on github (https://github.com/bbabayan/RL_beliefstate).

**Quantification and statistical analysis**. The values reported in the text and figures are the mean ± SEM. All data analysis was performed in MATLAB 2014b (Mathworks). Non-parametric tests were used where appropriate. When using parametric tests (t-test and ANOVA), we verified that data did not deviate significantly from a normal distribution, using a $\chi^2$ goodness-of-fit test. Tests were

two-tailed, except when otherwise mentioned, alpha was set at 0.05. Sample size was not predetermined.

**Data availability**. The data that support the findings of this study are available from the corresponding authors upon reasonable request.

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

## Acknowledgements

We thank members of the Gershman and Uchida labs for insightful discussions; S. Matias, Z. Mainen (Champalimaud Institute of Unknown), C. Burgess, M. Andermann (Harvard Medical School), and M.W. Mathis for advice on fiber photometry; Edward Soucy (CBS neuro-engineering platform) for instrumentation assistance; C. Dulac for sharing resources; and V. Jayaraman, R.A. Kerr, D.S. Kim, L.L. Looger, and K. Svoboda from the GENIE (Genetically-Encoded Neuronal Indicator and Effector) Project at the

Howard Hughes Medical Institute's Janelia Farm Research Campus for providing the AAV-GCaMP6f through the University of Pennsylvania Vector Core. This work was supported by the National Institutes of Health grants R01MH095953 (N.U.), R01MH101207 (N.U.), R01MH109177 (S.G.), Harvard Mind Brain and Behavior faculty grant (S.G. and N.U.), and Fondation pour la Recherche Medicale grant SPE20150331860 (B.B.).

## Author contributions

S.J.G. and B.M.B. designed the task, with help from N.U.; B.M.B. collected and analyzed data; B.M.B. and S.J.G. constructed the computational models; and B.M.B., N.U., and S.J.G. wrote the manuscript.

## Additional information

**Competing interests:** The authors declare no competing interests.

