## [Peer Review File · Nature Communications]

Reviewers' comments:

Reviewer #1 (Remarks to the Author):

Dopamine neurons in the midbrain ventral tegmental area are believed to encode reward prediction error (RPE) – they fire phasically to signal the difference between predicted reward and the currently experienced reward. The RPE then play a critical role in driving reinforcement learning. In this study, Babayan and colleagues introduced the concept of belief state and asked how it could be used to improve the calculation of RPE. Specifically, the authors propose that the belief state $b(s)$ is computed with Bayesian theorem: $b(s) = P(r|s) * P(s) / P(r)$. This value corresponds to the probability of being in a potential state, given previous reward information. Further, the RPE δ is calculated as $r - V(b)$, where r indicates received reward value and $V(b)$ correspond to the expected reward value at the belief state of (b) .

Similar question was addressed recently by a study on monkeys working on a visual decision task (2017, Current Biology, Lak et. al). Here, the authors tested the concept of belief state by recording the activity of VTA dopamine neurons using fiber photometry from mice that had been trained in a task with two potential states defined by a large reward and a small reward. Their model predicts non-monotonic RPE (in later trials) when a rare block of intermediate reward is randomly inserted between the blocks of large and small rewards. Their data fit quite well with the model for the first and second trials, but not for later trials. The authors conclude that state inference plays a critical role in reinforcement learning.

Overall, I like the concept of “belief state” and the approach of combining computational modeling with recordings from VTA dopamine neurons in behaving mice. However, I have several major concerns that the authors should address before publication.

Major Concerns:

1) Experiments. The experiments seem a bit rushed. The authors recorded from 11 mice. The Ca²⁺ indicator GCaMP was expressed either by mouse crossing or AAV viral transgene. The Ca²⁺ signals were recorded from either somata in the VTA or from the axonal terminals in the ventral striatum. The authors claimed that there was no major difference among individual mice. At the same time, data were normalized, largely because large variance. In addition, the inter-trial interval was only 4 s, which was so short that neuronal responses most likely could be affected by the activity in the previous trial. Both fiber photometry and classical conditioning behavior tests are very straightforward. I don't see any major difficulty in performing the experiments more rigorously.

2) How are Ca²⁺ signals related to neuronal spiking? Clearly, fiber photometry does not report basal activity well, although the authors know perfectly from their previous recordings that VTA neurons fire tonically at the basal state and phasically to reward-related signals. Therefore, the authors should at least discuss how their experimental data and the RPE value in the model are related to neuronal spiking.

3) I am pleased to see that the model fits well with the data for trials 1 and 2 following the rare introduction of intermediate reward after alternating small and large reward training (Figure 3b). However, the experimental data failed to obey RL model with belief states (Extended Data Figure 6), indicating that the model performed very poorly for trials 3-5. What is wrong here? Bad experiments, or bad theory? This must be resolved, otherwise the model seems to be quite limited.

4) In Fig 2, the authors show that the activity of dopamine neurons during the first two trials predicts mouse licking behavior during trials 2 and 3. If the change in anticipatory licks in trial 3 is truly

determined by dopamine response in trial 2, we should observe the following two phenomena: first, transient inhibition of dopamine neurons following 2 or 4 μL sucrose at trial 2 would decrease anticipatory licks in trial 3; second, transient activation of dopamine neurons following 6 or 8 μL sucrose at trial 2 would increase anticipatory licks at trial 3. It is unclear from the plot in Fig. 2f whether this is the case. The authors should also provide anticipatory lick data for trials 4 and 5, so that readers can better examine how the dopamine activity in the previous trial affects anticipatory licks in the following trial and how the behavior data fit with what is computed by the belief-state-based RL.

Minor Concerns:

- 1) It would be nice if the author choose a better example for Figure 2d. According to the current example of Figure 2d, the activity of dopamine neurons responding to 10 μL sucrose was smaller than that to 2 or 4 μL sucrose and no larger than that to 6 or 8 μL sucrose, inconsistent with the non-monotonic trend of grouped dopamine responses displayed in Figure 2e.
- 2) In Extended Data Figure 1, the scale range of RPE (δ) should be adjusted within -0.5 to 0.5 for the last column.
- 3) I am puzzled by the data that well-trained mice did not make anticipatorily lick during odor exposure in classic conditioning (Extended Data Figure 2a) but those mice trained with alternating blocks s1 and s2 showed robust anticipatory licking during odor presentation (Extended Data Figure 3a). Why?

Reviewer #2 (Remarks to the Author):

Babayan et al present a very elegant paper that suggests that dopamine (DA) release in the ventral striatum may be more complex than simply reflecting a RPE, predicted by traditional models of classical conditioning. The data indicate that DA activity is influenced by belief states and may signal reward values in non-linear manners.

I think this is an excellent finding, is solid, and fits well in Nature Communications.

I would like to ask the authors to revise the paper predominantly for clarity, to better relate their work to previous research of DA activity, and to reformat their paper to Nature Communications format (making it easier to read).

Main suggestions

-In the introduction, the authors state that the "states were sufficiently ambiguous". I don't understand why that is. If there is a reason, the authors need to clarify, if not, might they want to make this claim softer?. I agree that the main source of feedback about the "state" (or block) the mouse is in is the outcome size (e.g. the CS-odor only tells the timing of the outcome but not its size). However, after the first trial, for at least 5 trials (minimum block trial number) the mouse should not be confused about the state he is in, right? If this is wrong, what is a measure of ambiguity or what am I missing about the task design?

-The properties of DA neurons described by the authors are not entirely consistent with simple "arousal" explanations of the non-linearity they observe (which is great), and at first sight could appear closer to the way some people conceptualize "surprise". I suggest explicitly discussing this

issue. On a related note, DA neurons that project to dorsal striatum may have different properties (and may be closer to arousal/salience/surprise carrying units; Matsumoto and Hikosaka, 2010, Nature). Is it worth discussing that the data collected here are specific to ventral-striatum DA circuit? If anything, it is more exciting and leads to new questions about other types of DA neurons.

-I found the classical conditioning data more than just a form of verification. As the authors know, a large number of models of DA functions assume DA neurons display increases in activity that follow risky predictions which terminate at outcome time. These increases are theoretically selective for risk and have been only shown in one single paper which has been difficult to replicate. Models have been proposed to explain this (e.g. Dayan's idea that Bergman's trace conditioning tasks fail to show ramping due to slow learning rates in trace; etc). However, evidence is mounting that most or many DA neurons do not ramp to risky outcomes. It appears that the classical conditioning data here nicely replicates those impressions. I think it's worth explicitly discussing this issue here because the authors have a clear "identified DA population signal" to ventral striatum.

-A recent Takahashi paper (Neuron, September, 2017) shows that DA signals the sensory features of rewards. It is worth discussing the relationship of those findings to yours.

-Would it be helpful to show block-start tone related responses of DA? I suspect there may be no difference in DA activation between start of small reward block and large reward block (though they alternate and mice should be able to predict which block he will be in?). Is that true? Have the authors considered the meanings of this type of finding (or other types of block-start effects)? A previous paper (Bromberg-Martin, 2010, Neuron) analyzed trial start activity and was able to show a more complex anticipatory role of DA neurons as animals (macaques) expected belief and value state changes (from long reward to long punishment states/blocks). Of note, those data seem highly relevant here.

-I don't have experience with fiber photometry. So, one question that may be easy to answer but that arose in my reading of your paper (and that I have no online experience with) is whether the baseline of DA activity changes as the trials continue in a given block? This may be in Supplemental, or elsewhere, but I missed it. I think that would be worth reporting and on a very related note, more details about how you "normalize" (e.g. min-max) would be helpful as well.

Minor concerns addressing style and clarity

-Figure 1 – the authors state "then intermediate rewards are introduced". This is a totally vague description of the crucial manipulation and does not give the clever design any credit. Please elaborate and describe the task in great detail in the legend (within the word limits of Nat. Com).

-Extended data Figures 2-3 are very important figures. I would suggest moving many of the panels from them to the main body of the manuscript.

-Sentence on line 34 needs to be elaborated for clarity. It is a key point to the authors.

-Line 55 of introduction – I would front load this so that the rest is easy and exciting to read. It may increase the "word count" but in Nature Communications this is less of an issue.

We would like to thank the referees for their insightful and constructive comments. We have performed additional analyses and addressed all concerns.

In the following, the reviewers' comments are in slab-serif font (*Courier New*), our response appears in sans-serif font (*Arial*) and text from the revised manuscript and previous publications appear in *italic*.

Reviewer #1 (Remarks to the Author)

Dopamine neurons in the midbrain ventral tegmental area are believed to encode reward prediction error (RPE) - they fire phasically to signal the difference between predicted reward and the currently experienced reward. The RPE then play a critical role in driving reinforcement learning. In this study, Babayan and colleagues introduced the concept of belief state and asked how it could be used to improve the calculation of RPE. Specifically, the authors propose that the belief state $b(s)$ is computed with Bayesian theorem: $b(s) = P(r|s) * P(s) / P(r)$. This value corresponds to the probability of being in a potential state, given previous reward information. Further, the RPE δ is calculated as $r - V(b)$, where r indicates received reward value and $V(b)$ correspond to the expected reward value at the belief state of (b) .

Similar question was addressed recently by a study on monkeys working on a visual decision task (2017, *Current Biology*, Lak et. al). Here, the authors tested the concept of belief state by recording the activity of VTA dopamine neurons using fiber photometry from mice that had been trained in a task with two potential states defined by a large reward and a small reward. Their model predicts non-monotonic RPE (in later trials) when a rare block of intermediate reward is randomly inserted between the blocks of large and small rewards. Their data fit quite well with the model for the first and second trials, but not for later trials. The authors conclude that state inference plays a critical role in reinforcement learning.

Overall, I like the concept of "belief state" and the approach of combining computational modeling with recordings from VTA dopamine neurons in behaving mice. However, I have several major concerns that the authors should address before publication.

Major Concerns:

1) Experiments. The experiments seem a bit rushed. The authors recorded from 11 mice. The Ca²⁺ indicator GCaMP was expressed either by mouse crossing or AAV viral transgene. The Ca²⁺ signals were recorded from either somata in the VTA or from the axonal terminals in the ventral striatum. The authors claimed that there was no major difference among individual mice. At the same time, data were normalized, largely because large variance. In addition, the inter-trial interval was only 4 s, which was so short that neuronal responses most likely could be affected by the activity in the previous trial. Both fiber photometry and classical conditioning behavior tests are very straightforward. I don't see any major difficulty in performing the experiments more rigorously.

Response: We apologize that the experiments seemed rushed. We would like to first mention that our experiments require significant amount of time and effort for each animal. Although mice can be trained in simple odor-outcome association paradigms in several days (e.g. our previous studies, Cohen et al., 2012; Menegas et al., 2017), we spent a significantly longer time to train mice in the present task (> 20 days of training on the reference small and big blocks) in order to ensure that each animal develops discrete belief states. Furthermore, our analysis relies on rare 'probe' trials consisting of the presentation of intermediate reward sizes. Thus, the data collection took more than 20 days for each animal. This meant that we had to perform the behavioral training and experiment every day for more than 40 days for each mouse. We agree that fiber photometry is relatively straightforward. Nonetheless, this unique experimental design made the performance of the experiments very time-consuming. Because the data looked consistent across different experimental conditions (we discuss this below), we thought that pooling the data is well justified, and we decided that we can make conclusions based on the collected data, and that performing more experiments is not justified due to the costs with regard to the experimenter and a further sacrifice of animals.

We apologize that our justification of pooling and normalization was not sufficient. To address these concerns, we now discuss these issues in Results (page X), and present the data from different conditions (Figure R1 below, now Supplementary Figure 5). We recorded from 11 mice that are divided into the following conditions:

1. Mice expressing GCaMP6f transgenetically in DAT-positive neurons and recorded from VTA cell bodies (n = 5)
2. Mice expressing GCaMP6f through a viral construct in DAT-positive neurons and recorded from VTA cell bodies (n = 2)

3. Mice expressing GCaMP6f through a viral construct in DAT-positive neurons and recorded from dopamine neuron terminals in the ventral striatum (n = 4).

In Figure R1, the upper row (a - c) shows the average across mice within each recording condition. The monotonicity and non-monotonicity of the responses in trials 1 and 2, respectively, are observed in each recording condition (a - c). We used these data to justify pooling across recording conditions.

Figure R1. Dopamine responses on trials 1 and 2 plotted separately based on recording conditions. (a, d) Dopamine responses for mice expressing transgenetically GCaMP6f in DAT-positive neurons and recorded from VTA cell bodies (n = 5). (b, e) Dopamine responses for mice expressing GCaMP6f through a viral construct in DAT-positive neurons and recorded from VTA cell bodies (n = 2). (c, f) Dopamine responses for mice expressing GCaMP6f through a viral construct in DAT-positive neurons and recorded from dopamine neuron terminals in the ventral striatum (n = 4). The upper row (a - c) shows the average across mice, while the lower row (d - f) shows the same average after normalizing within mice through min-max normalization using trial 1's response as reference for the minimum and maximum values. This normalization corrects for the different amplitudes in GCaMP signals across the different recording conditions, but preserves the features observed in each recording condition. Note that the monotonicity and non-monotonicity of the responses in trials 1 and 2, respectively, are observed in each recording condition (a - c). Data represents mean ± s.e.m.

In addition to the consistency, these plots also show that the amplitude of GCaMP signals varied across the different recording conditions. This is largely due to lower expression levels of GCaMP in transgenic mice compared to those with viral expression, resulting in the overall smaller signals in transgenic mice. There was also some variability in signal intensity across animals within each recording condition. This is likely due to different expression levels of GCaMP and fiber locations. Therefore, for illustration purposes, we chose to normalize the signals from each individual mouse to plot the summary data in one plot (as shown in Figures 2 and 3). The normalization we used was a min-max normalization ($y = (x - \min) / (\max - \min)$) to rescale the GCaMP signals in the 0 to 1 range. In our original manuscript, we normalized the data for each

trial condition separately. Following the Reviewer's third point (detailed below), we now notice that this normalization was misleading since each trial was rescaled with reference to itself, hence masking any change of amplitude across trials. We now propose to normalize within mice using trial 1's response as reference for the minimum and maximum values for the min-max normalization (i.e. $y = (x - \min_{\text{trial1}}) / (\max_{\text{trial1}} - \min_{\text{trial1}})$). The lower row (d - f) shows the average after performing this normalization with each mouse's signal. We believe that this normalization corrects for the differences in amplitude across mice, while preserving the features observed in each recording condition, i.e. monotonicity in trial 1 and non-monotonicity in trial 2. It is also important to note that the models were not fit on the normalized data. We now explicitly explain these points in Results and Methods:

Results section, page 7, lines 125-138:

'These monotonic and non-monotonic patterns on trials 1 and 2, respectively, were observed in our three different recording conditions: (1) in mice expressing GCaMP6f transgenetically in DAT-positive neurons and recorded from VTA cell bodies (n = 5), (2) in mice expressing GCaMP6f through a viral construct in DAT-positive neurons and recorded from VTA cell bodies (n = 2); (3) in mice expressing GCaMP6f through a viral construct in DAT-positive neurons and recorded from dopamine neuron terminals in the ventral striatum (n = 4) (Supplementary Fig. 5, a - c). Although these patterns were observed in each condition, the amplitude of the signal varied across the different recording conditions, largely due to lower expression levels of GCaMP in transgenic mice compared to those with viral expression and overall variability in signal intensity across animals within each recording condition. Therefore, for illustration purposes, we normalized the signals from each individual mouse using trial 1's response as reference for the minimum and maximum values for the min-max normalization ($y = (x - \min_{\text{trial1}}) / (\max_{\text{trial1}} - \min_{\text{trial1}})$) to rescale the GCaMP signals in the 0 to 1 range (Supplementary Fig. 5, d - f, Fig. 3, Fig. 4).'

Methods section, page 19, line 388-395:

'Since the absolute level of fluorescence was variable across mice that expressed GCaMP6f through viral injection or transgenetically (Supplementary Figure 4), for illustration purposes to summarize the data in one plot, each mouse's mean US response across rewards was normalized by min-max normalization when pooled together. The normalization was performed within each mouse, using the given mouse's trial 1 response as reference for the minimum and maximum values for the min-max normalization such that $y = (x - \min_{\text{trial1}}) / (\max_{\text{trial1}} - \min_{\text{trial1}})$. Of note, the models were not fit on the normalized data.'

One of the reasons that we recorded from these conditions is that because the present study was one of the first in our lab that used fiber photometry, we wanted to test

whether we observe different signals between these recording conditions. Although we agree that these conditions could have been designed in a more systematic way, our initial data indicated that the main results were similar across these conditions. Furthermore, as mentioned above, our experimental design required us to perform >40 days of daily behavioral training and data collection. Taking these into account, we decided that rather than comparing these recording conditions systematically (e.g. so that we can discuss potentially interesting differences with respect to signal sources), we rather pooled data to address our main question.

Finally, we would like to clarify the inter-trial interval duration. It was on average 8.7 s, with a fixed initial period of 4 s and then an additional interval drawn from an exponential distribution of mean 4.7 s. The initial fixed period allowed no ITI to be shorter than 6.8 s to allow GCaMP signals to go back to baseline between trials and the flat hazard function ensured mice had constant expectation of when the next trial would begin.

We have clarified this point in Methods (Page 16, Lines 322-326):

‘Inter-trial intervals were on average 8.7s, composed of an initial fixed 4 s period, to ensure GCaMP signals went down to baseline between trials, followed by an interval drawn from an exponential distribution (mean: 4.7 s), resulting in a flat hazard function such that mice had constant expectation of when the next trial would begin. Mice did 30 blocks per day (150 trials).’

2) How are Ca²⁺ signals related to neuronal spiking? Clearly, fiber photometry does not report basal activity well, although the authors know perfectly from their previous recordings that VTA neurons fire tonically at the basal state and phasically to reward-related signals. Therefore, the authors should at least discuss how their experimental data and the RPE value in the model are related to neuronal spiking.

Response: Thank you for pointing this out. We have explored the relationship between spikes and photometry signals in multiple ways. First, photometry signals that we obtained in a classical conditioning paradigm (e.g. Supplementary Fig. 2 and the data in Menegas et al., 2017) were similar to our electrophysiological recording data that we observed in similar task conditions (e.g. Cohen et al., 2012, Eshel et al., 2015, Tian et al., 2015). Second, in an unpublished work (Menegas et al., submitted), we have obtained a ‘dose-response’ curve of reward responses in dopamine terminals in ventral striatum using photometry. The result showed that reward response increases

monotonically as a function of reward size (Figure R2, a and b), consistent with what we obtained in electrophysiological recording (Figure R2, c, reproduced from Eshel et al., 2015). This result supports relatively linear relationship between spikes and photometry signals. Third, our previous results also indicated that our photometry system has the ability to monitor a transient decrease from the baseline activity (e.g. when an expected reward is omitted) (Menegas et al., 2017).

Figure R2. Electrophysiological recordings of dopamine neurons to unexpected reward delivery. Quantifications of responses from electrophysiological recordings of light-identified dopamine neurons (reproduced from Eshel et al., 2015). Data represent mean \pm s.e.m. [Redacted]

As the reviewer pointed out, it is less clear whether we can monitor long-timescale changes in baseline activity due to technical limitations such as bleaching of the calcium indicator. In addition, a majority of previous work studying RPEs focused on phasic responses. Because of these reasons, the present study focused on phasic responses. Now we discuss this limitation and our motivation to focus on phasic responses in Results and Methods.

Of note, although there are some studies that indicated changes in baseline firing (in particular, those that monitored activity in slice or anesthetized animals), there are fewer results in awake behaving animals. Our previous study (Cohen et al., 2016) showed that the baseline firing of VTA dopamine neurons was not altered according to the slow-timescale overall value (blocks of reward trials versus air puff trials) in behaving mice.

Results section, page 5, lines 76-82:

‘We focused our analysis on the phasic responses. Indeed, calcium imaging limits our ability to monitor long-timescale changes in baseline due to technical limitations such as bleaching of the calcium indicator, moreover a majority of previous work studying dopamine neurons has shown reward prediction error-like signalling in the phasic

responses^{1,3,11}. Similarly to single cell recordings^{1,3,11}, population activity of dopamine neurons measured by fiber photometry in the VTA¹⁹ (Supplementary Fig. 2) or in terminals of dopamine neurons projecting to the ventral striatum^{15,20} show canonical RPE coding in classical conditioning tasks.'

Methods section, page 18, line 366-369:

'For GCaMP activity, we focused our analysis on the phasic responses. Indeed, a majority of previous work has shown reward prediction error-like signaling in the phasic responses of dopamine neurons and technical limitations such as bleaching limit our ability to monitor long-timescale changes in baseline using calcium imaging. Fluorescence data was acquired at 1 kHz.'

Finally, the exact sources of calcium signals remain unclear. Most, if not all, of *in vivo* calcium imaging studies assume that large calcium influxes through voltage-gated calcium channels evoked by spikes dominate calcium signals that they measure. Nonetheless, this might not be true in some systems. With respect to the dopamine system, there are some unique points that need to be taken into account when we interpret calcium imaging data. First, dopamine neurons have a mechanism to maintain the baseline, pace-making activity which relies on calcium. Second, increasing evidence suggests that dopamine release is regulated at the level of axon terminals, through cholinergic and glutamatergic mechanisms (e.g. Threlfell et al., 2012). We now realize that our data using multiple recording conditions may speak to this issue: because our main results hold whether we monitored the activity from cell bodies or axons of dopamine neurons, these additional processes are unlikely to affect our main conclusions.

We now discuss these points in the discussion (page 12, lines 251-262):

'The exact sources of calcium signals remain unclear. Most, if not all, of in vivo calcium imaging studies assume that large calcium influxes through voltage-gated calcium channels evoked by spikes dominate calcium signals that they measure. Nonetheless, this might not be true in some systems. With respect to the dopamine system, there are some unique points that need to be taken into account when we interpret calcium imaging data. First, dopamine neurons have a mechanism to maintain the baseline, pace-making activity which relies on calcium³⁴. Second, increasing evidence suggests that dopamine release is regulated at the level of axon terminals, through cholinergic and glutamatergic mechanisms³⁵⁻³⁷. Furthermore, cholinergic interneurons in the dorsomedial striatum have been shown to track beliefs about current state³⁸. However, because our main results hold whether we monitored the activity from cell bodies or axons of dopamine neurons,

these additional processes are unlikely to affect our observation of state inference modulation of dopamine neuron activity.'

3) I am pleased to see that the model fits well with the data for trials 1 and 2 following the rare introduction of intermediate reward after alternating small and large reward training (Figure 3b). However, the experimental data failed to obey RL model with belief states (Extended Data Figure 6), indicating that the model performed very poorly for trials 3-5. What is wrong here? Bad experiments, or bad theory? This must be resolved, otherwise the model seems to be quite limited.

Response: Thank you for pointing this out. Indeed, Supplementary Figure 7c (previously Extended Data Figure 6) appears to indicate that the model fits get worse in trials 3-5. However, this is, at least in part, due to our normalization procedure: in the original manuscript, we normalized the data for each trial condition separately. We now notice that this normalization was misleading. In our original manuscript, normalizing the data for each trial separately resulted in each trial being rescaled with reference to itself, hence masking any change of amplitude across trials. We now propose to normalize within mice using trial 1's response as reference for the minimum and maximum values for the min-max normalization (i.e. $y = (x - \min_{\text{trial1}}) / (\max_{\text{trial1}} - \min_{\text{trial1}})$). This normalization results in the following updated panel c for Supplementary Figure 7:

Using the same normalization across trials (Trial 1 – 5) shows shallower responses in later trials. The reinforcement learning model with belief state also predicts a flattening of the non-monotonic reward prediction error pattern with increasing exposure to the same reward. In Figure R3a, we generated reward prediction errors across trials using the average parameters obtained across mice (Supplementary Table 1) to illustrate how these signals change across trials for the two reinforcement learning models, without and with belief state. In panel b of Figure R3, we computed the sum of squared errors

between both simulations for all trials. It was maximal for Trial 2. That is, the two models made most different predictions in Trial 2. This is why we focused our analysis on trial 2.

Figure R3: Reward prediction errors across trials. (a) Standard and belief state reinforcement learning models were simulated using the average parameters across mice (**Supplementary Table 1**). (b) Sum of squared errors between simulations from both models. Trial 2 shows the strongest difference.

The dopamine data presented in Supplementary Figure 7a show shallower non-monotonicity across trials. It is, however, interesting to note that different mice show the non-monotonic reward response modulation at varying degrees on distinct trials. For example, in Figure R4, Mouse 4 shows a strong non-monotonic pattern on trial 2, which then becomes shallower on the following trials, whereas Mouse 9 shows a more sustained non-monotonic pattern across trials 2 to 5.

Figure R4. Examples of individual dopamine responses. Data represents mean \pm s.e.m.

While we focused the main results on trial 2, which is the trial with a strongest belief state modulation (Figure R3), the non-monotonic pattern is not necessarily restricted to it. For this reason, we fitted the models to all trials (1 to 5) for each mouse individually,

accounting for potentially different learning rates or different sensory variances across mice, which could affect the dynamics of reward prediction error across trials. The quality of fit comparisons in Supplementary Table 1 is for all trials. Across all trials, the reinforcement learning model with belief state explains the data better.

We would like to thank the reviewer. We believe that the current normalization greatly improved our presentation the result (model fit). We believe that this now shows that our model is not very limited. Here, we aimed at providing the most parsimonious explanation for all of our data, using a minimal number of parameters and of *a priori* assumptions in the model. This approach may appear to produce fits that are not perfectly capturing our recording data, yet we do believe that it addresses our question of an influence of state inference of dopaminergic RPE, since even a more enhanced standard reinforcement learning (as here with values depending on block history) cannot account for the pattern we measure across all trials, in all mice.

In the manuscript, we have made clearer the prediction of the evolution of the RPE across trials in the results section, and have expanded Supplementary Figure 7 for it to include Figures R3 and R4.

Results section, page 8, lines 141-152:

'The non-monotonic pattern observed on trial 2 was consistent with our hypothesis of belief state influence on dopamine reward RPE (Fig. 1d). We focused our analysis on trial 2 since, according to our model, that is the most likely trial to show an effect of state inference with the strongest difference from standard RL reward prediction errors (Supplementary Fig. 6a, b). Both reinforcement learning models predict weaker prediction error modulation with increasing exposure to the same reward and we observed weaker versions of this non-monotonic pattern in later trials (Supplementary Fig. 7c and Supplementary Fig. 8a). It is however interesting to note that different mice showed a non-monotonic reward response modulation at varying degrees on distinct trials. For example, Mouse 4 showed a strong non-monotonic pattern on trial 2, which then became shallower on the following trials, whereas Mouse 9 showed a more sustained non-monotonic pattern across trials 2 to 5 (Supplementary Fig. 7d).'

Corrected and expanded Supplementary Figure 7:

Supplementary Figure 7. Dopamine reward responses and model fits across trials. **a** Standard and belief state reinforcement learning models were simulated using the average parameters across mice (Supplementary Table 1). **b** Sum of squared errors between simulations from both models. Trial 2 shows the strongest difference. **c** Normalized dopamine responses to rewards and model fits to dopamine responses of the RL models without or with belief states, with two free initial values or priors. **d** Examples of individual dopamine responses. Data represents mean \pm s.e.m.

4) In Fig 2, the authors show that the activity of dopamine neurons during the first two trials predicts mouse licking behavior during trials 2 and 3. If the change in anticipatory licks in trial 3 is truly determined by dopamine response in trial 2, we should observe the following two phenomena: first, transient inhibition of dopamine neurons following 2 or 4 μ L

sucrose at trial 2 would decrease anticipatory licks in trial 3; second, transient activation of dopamine neurons following 6 or 8 μL sucrose at trial 2 would increase anticipatory licks at trial 3. It is unclear from the plot in Fig. 2f whether this is the case. The authors should also provide anticipatory lick data for trials 4 and 5, so that readers can better examine how the dopamine activity in the previous trial affects anticipatory licks in the following trial and how the behavior data fit with what is computed by the belief-state-based RL.

Response: The Reviewer’s intuition about the relation between dopamine response and the subsequent change in anticipatory licking is correct. As mentioned in the manuscript, and illustrated on the first plot of the figure below, there is a weak but significant positive correlation between dopamine activity and lick change on the following trial across all trials (Pearson’s $r = 0.12$, $P = 1.0 \times 10^{-37}$, Figure R5, left). When splitting the data for each trial type, the relationship still holds, with inhibition or lower activations of dopamine neurons often followed by a decrease in anticipatory licking whereas transient activations of dopamine neurons tended to be followed by increased anticipatory licking (Figure R5, right).

Figure R5. Correlation analysis between dopamine neuron activity on trial t and lick rate change from trial t to trial $t+1$ within blocks. Each point represents an individual trial.

Although at the individual trial level, significant positive correlations exist between dopamine activity and subsequent change in licking rate, the average change in anticipatory licking for all trial transitions across mice visibly follows the pattern of dopamine activity for the first two transitions (Figure R6).

Figure R6. Dopamine neuron activity and differential anticipatory licking within blocks. Data represent mean \pm s.e.m. $n = 11$

In the manuscript, we have made the relationship between differential anticipatory licking and dopamine responses clearer and have included an additional Supplementary Figure (Supp. Fig. 9) including Figures R6 and R7.

Results section, page 8, lines 160-174:

*‘From trial 1 to trial 2, mice changed their anticipatory licking proportionally to the volume (Fig. 3d) but showed a non-monotonic change from trial 2 to trial 3 (Fig. 3h; highest adjusted r^2 for a cubic polynomial fit, Supplementary Fig. 6d). Fits of linear and polynomial functions to the change in anticipatory licking revealed highest adjusted r^2 for cubic polynomial fits for both transitions from trial 1 and 2 (Supplementary Fig. 6c), although the linear fit still provided a decent fit (adjusted $r^2 = 0.94$). Thus, dopamine activity and change in anticipatory licking both showed modulation according to our prediction of the influence of belief state on RPE (Fig. 1d). **Although the average change in anticipatory licking for transitions from trial 3 to 5 did not seem to visibly follow the pattern of dopamine activity (Supplementary Fig. 9a), a trial-by-trial basis showed that dopamine responses on reward presentation were significantly correlated with a change of licking on following trial for all trial transitions within blocks (trial 1 to 5, Pearson’s r , $p < 2.5 \times 10^{-3}$, Supplementary Fig. 9b), suggesting that inhibition or lower***

activations of dopamine neurons were more often followed by a decrease in anticipatory licking whereas transient activations of dopamine neurons tended to be followed by increased anticipatory licking.'

Corrected and expanded Supplementary Figure 9:

Supplementary Figure 9. Dopamine neuron activity and differential anticipatory licking within blocks. a Dopamine neuron activity on reward delivery for trials 2 to 5 (top) and corresponding differential lick rate (bottom). Data represent mean \pm s.e.m. **b** Correlation analysis between dopamine neuron activity on trial t and lick rate change from trial t to trial $t+1$ within blocks. Each point represents an individual trial. $n = 11$

Minor Concerns:

1) It would be nice if the author choose a better example for Figure 2d. According to the current example of Figure 2d, the activity of dopamine neurons responding to 10 μL sucrose was smaller than that to 2 or 4 μL sucrose and no larger than that to 6 or 8 μL sucrose, inconsistent with the non-monotonic trend of grouped dopamine responses displayed in Figure 2e.

Response: We agree that the presented figures are counter-intuitive, at least, at a glance. Contrary to the reviewer's suggestion, however, the data in Figure 3d produces a non-monotonic change in the photometry data (reproduced in Figure R7). This is because GCaMP responses display complex patterns to reward, with a specific combination of peak activity and response dynamic. For example, in the data presented in Figure 3 (and reproduced below, Fig. R7), a presentation of 1 μL reward in Trial 1 triggers a biphasic response with an initial peak and then a dip below baseline. Because of these dynamics, we chose to quantify GCaMP responses as the average activity over 1 second following reward presentation (equivalent to the area under the curve; indicated by the horizontal black bar). This quantification shows that in the example mouse shown in Figure 3, the response to 10 μL is smaller than that to 2 or 4 μL and larger than that to 6 or 8 μL . This result, thus, shows a representative pattern consistent with the population average.

Figure R7. Quantification of the activity of the mouse shown in Figure 2. GCaMP dF/F responses are quantified as the mean response after reward presentation (0–1 s, indicated by a solid black line in a and c). Data represents mean \pm s.e.m.

Because the relationship between the example and the population is likely to be counter-intuitive, we now include the plot of dF/F plotted against reward sizes from the example animal in the main figure (Figure 3).

We have also corrected the figure legend to explain the quantification method:

‘On trial 1, dopamine neurons show a monotonically increasing response to increasing rewards (a, individual example), quantified as the mean response after reward presentation (0–1 s, indicated by a solid black line in a) in the individual example (b) and across mice (c).’

Corrected Figure 3:

Figure 3. Dopaminergic and behavioural signature of belief states. **a - c** Dopamine neurons activity on trial 1. Dopamine neurons show a monotonically increasing response to increasing rewards (a, individual example), quantified as the mean response after reward presentation (0–1 s, indicated by a solid black line in a) in the individual example (b) and across mice (c). **d** Change in anticipatory licking from trial 1 to trial 2. Mice increase their anticipatory licking after trial 1 proportionally to the increasing rewards. **e - g** Dopamine neurons activity on trial 2. Dopamine neurons show a non-monotonic response pattern to increasing rewards (e and f, individual example), quantified across all mice (g). **h** Change in anticipatory licking from trial 2 to trial 3. Whereas mice do not additionally adapt their licking for the known trained volumes (1 and 10 μL) after trial 2, they increase anticipatory licking for small intermediate rewards and decrease it for larger intermediate rewards in a pattern, which follows our prediction of belief state influence on RPE. $n = 11$, data represent mean \pm s.e.m.

2) In Extended Data Figure 1, the scale range of RPE (δ) should be adjusted within -0.5 to 0.5 for the last column.

Response: Thank you for pointing this out. In Supplementary Figure 1, the 4th column shows the theoretical RPE, which is centered around 0 and within the -0.5 to 0.5 range. The last column shows the theoretical RPE fitted to the GCaMP responses through linear regression. This regression accounted for the fact that in our task most reward

responses were positive. This is likely due to temporal uncertainty (Fiorillo et al., 2008; Kobayashi and Schultz, 2008) and has been observed universally in our previous work (e.g. Cohen et al., 2012; Eshel et al., 2015). Hence the range of these fitted RPEs is not centered around 0, but rather vary between 0 and 1. We have clarified this point in the legend:

‘Supplementary Figure 1. RL models tested. *RL models tested. Six model variants were tested. For each model, from left to right, the model's state space is represented, followed by the delivered reward (r), which is compared to the expectation (value V of the state or belief state) to compute the RPE (δ) (a-f). The 4th column shows the theoretical RPE, which is centered around 0. The last column shows the theoretical RPE fitted to the GCaMP responses through linear regression. The main distinction between the standard RL models and the belief state models is the state representation, with a single state in the case of the standard RL model due to the ambiguity of the odor. The last two columns (g-l) show the theoretical value and RPE on trial 2, obtained by fitting each model's RPE to the GCaMP responses using linear regression. This regression accounted for the fact that in our task most reward responses were positive, which is likely due to temporal uncertainty (Fiorillo et al., 2008; Kobayashi and Schultz, 2008).’*

3) I am puzzled by the data that well-trained mice did not make anticipatorily lick during odor exposure in classic conditioning (Extended Data Figure 2a) but those mice trained with alternating blocks s1 and s2 showed robust anticipatory licking during odor presentation (Extended Data Figure 3a). Why?

Response: Supplementary Figure 2a shows the presentation of an unpredicted reward. Since no odor or any other cue predicted the delivery of water, the mouse started licking only when it detected the water. On the other hand, Panels b and c show anticipatory licks for odors predicting 90% and 50% reward probabilities, which is in line with the data from the mice trained on the belief state task presented in Figure 2 (previously Extended Data Figure 3a).

We clarified this point in the legend:

‘We presented 3 odours, which predicted the delivery of water one second later with either 90% (red), 50% (green) or 0% (black) probability. Unpredicted water was delivered on 10% of trials. On unpredicted water delivery trials, the mouse licked on water delivery (a). For odours predicting reward with 90% or 50% probability, the mouse showed anticipatory licking after odor presentation proportional to the probability of reward delivery (b, c).’

Reviewer #2 (Remarks to the Author):

Babayan et al present a very elegant paper that suggests that dopamine (DA) release in the ventral striatum may be more complex than simply reflecting a RPE, predicted by traditional models of classical conditioning. The data indicate that DA activity is influenced by belief states and may signal reward values in non-linear manners.

I think this is an excellent finding, is solid, and fits well in Nature Communications.

I would like to ask the authors to revise the paper predominantly for clarity, to better relate their work to previous research of DA activity, and to reformat their paper to Nature Communications format (making it easier to read).

Response: We thank the reviewer for the kind remark. We have reformatted the paper following the *Nature Communications* format (e.g. adding section headings, expanding the introduction) and follow up with detailed responses to the suggestions.

Main suggestions

-In the introduction, the authors state that the "states were sufficiently ambiguous". I don't understand why that is. If there is a reason, the authors need to clarify, if not, might they want to make this claim softer?. I agree that the main source of feedback about the "state" (or block) the mouse is in is the outcome size (e.g. the CS-odor only tells the timing of the outcome but not its size). However, after the first trial, for at least 5 trials (minimum block trial number) the mouse should not be confused about the state he is in, right? If this is wrong, what is a measure of ambiguity or what am I missing about the task design?

Response: Blocks are structured such that they all begin with a two seconds long sound, followed by five identical trials, each starting with the presentation of a unique odor followed by reward one second later. Indeed, only reward amount distinguishes the blocks, and this amount is stable within a block. Thus, as mentioned by the Reviewer, the task is designed such that within blocks, this ambiguity is lifted after the first trial.

The purpose of the twenty days of training on the reference blocks (s_1 and s_2) is for mice to learn the underlying structure of the task. We believe that mice do learn this structure since for the reference blocks (s_1 and s_2), both anticipatory licking and dopamine activity are stable from trials 2 to 5 (see Fig. 2 and figure R8 below). It is thus likely that within the reference blocks, the ambiguity disappears. However for the intermediate rewards, both anticipatory licking and dopamine activity continue changing across trials (see figure below). This continued change could indicate uncertainty in the amount of expected reward, even after trials 1 and 2. Indeed, when considering the mice's overall experience (with reference blocks and intermediate reward size blocks), the sound and the odor are associated with distinct reward sizes. They themselves are thus insufficiently informative of the block identity, making them inherently ambiguous. This ambiguity, maximal on trial 1, is gradually resolved within each block, seemingly at different rates for the reference and intermediate reward blocks.

Figure R8. Anticipatory licking and dopamine change across trials 2 to 5. Differential anticipatory licking and GCaMP dF/F responses are quantified as the change between two consecutive trials. The dotted vertical lines separates the reference blocks (s_1 and s_2) from the intermediate reward blocks. Data represents mean \pm s.e.m.

This inherent ambiguity is what we referred to as ‘sufficiently ambiguous’. We have corrected that part of the manuscript to make our reasoning clearer.

Results section, Page 3, Lines 45-54:

‘We designed a task that allowed us to test distinct theoretical hypotheses about dopamine responses with or without state inference. We trained 11 mice on a Pavlovian conditioning task with two “states” distinguished only by their rewards: an identical odor cue predicted the delivery of either a small (s_1) or a big (s_2) reward (10% sucrose water) (Fig. 1a). The different trial types were presented in randomly alternating blocks of five identical trials, and a tone indicated block start. Only one odor and one sound cue were used for all blocks, making the two states perceptually similar prior to reward delivery. This task feature resulted in ambiguous sound and odor cues, since they were

themselves insufficiently informative of the block identity, rendering the two states ambiguous with respect to their identity. This feature increased the likelihood of mice relying on probabilistic state inference.'

-The properties of DA neurons described by the authors are not entirely consistent with simple "arousal" explanations of the non-linearity they observe (which is great), and at first sight could appear closer to the way some people conceptualize "surprise". I suggest explicitly discussing this issue. On a related note, DA neurons that project to dorsal striatum may have different properties (and may be closer to arousal/salience/surprise carrying units; Matsumoto and Hikosaka, 2010, Nature). Is it worth discussing that the data collected here are specific to ventral-striatum DA circuit? If anything, it is more exciting and leads to new questions about other types of DA neurons.

Response: We appreciate the Reviewer's suggestion for providing interesting speculations regarding the nature of the dopamine signals we observed. In general, a 'prediction error' signal can be seen as a 'surprise' signal in the sense that it represents the deviation from expectation. One way to distinguish different surprise signals is to consider along what axis a surprise signal manifests. A reward prediction error is computed along the axis of value, and this is what we are considering in the present study.

As the Reviewer pointed out, increasing evidence suggests that dopamine neurons that project to the dorsal striatum signal different types of 'surprise' signal. In addition to the seminal work by Matsumoto and Hikosaka (2010), recent work from our lab (Menegas et al., 2017) and other labs (Lerner et al., 2016) indicated that dopamine neurons projecting to specific regions of the dorsal striatum are activated by both rewarding and aversive stimuli, consistent with the coding of 'salience'. Following the Reviewers' comment, we now discuss that (1) we have recorded from the canonical dopamine system that encode value prediction errors, and (2) the dorsal striatum may receive different types of dopamine signals, which warrant future investigations.

Discussion section, Page 12, Lines 243-249:

'Increasing evidence suggests that dopamine neurons that project to the dorsal striatum signal different types of signals. Indeed dopamine neurons projecting to specific regions of the dorsal striatum have been shown to be activated by rewarding, aversive and novel

stimuli^{15,30,31}. Here we recorded from the canonical dopamine system, involving VTA to ventral striatum loops, which encode value prediction errors. Whether other dopamine inputs projecting to other areas of the dorsal striatum and broadcasting different types of signals can also be modulated by inferred states remains to be addressed.'

-I found the classical conditioning data more than just a form of verification. As the authors know, a large number of models of DA functions assume DA neurons display increases in activity that follow risky predictions which terminate at outcome time. These increases are theoretically selective for risk and have been only shown in one single paper which has been difficult to replicate. Models have been proposed to explain this (e.g. Dayan's idea that Bergman's trace conditioning tasks fail to show ramping due to slow learning rates in trace; etc). However, evidence is mounting that most or many DA neurons do not ramp to risky outcomes. It appears that the classical conditioning data here nicely replicates those impressions. I think it's worth explicitly discussing this issue here because the authors have a clear "identified DA population signal" to ventral striatum.

Response: Thank you very much for appreciating the control data. We agree that whether dopamine neurons exhibit 'ramping' activity according to uncertainty is a controversial issue. Indeed, ramping signals have not been observed in our previous studies using electrophysiological recording from optogenetically-identified dopamine neurons (Tian et al., 2015). Fiber photometry data further extend these results, and will be important as it provides a population level result with a defined cell type and projection target. Although we very much agree that this line of research should be highlighted, we feel that this is out of the scope of the present study. We are planning to pursue this question more thoroughly in the future with a systematic investigation of different projection targets.

-A recent Takahashi paper (Neuron, September, 2017) shows that DA signals the sensory features of rewards. It is worth discussing the relationship of those findings to yours.

Response: Thank you for pointing this out. This is an important study. However, the present study does not address the issue of sensory features. We now discuss the distinction from our study in the discussion section, page 13.

‘A recent study found that dopamine neurons alter their responses based on changes in sensory features of reward⁵⁰. In the present study, we focused on reward prediction errors based on reward sizes. It is interesting to extend the present study using different sensory features (e.g. taste or smell of reward) that may define ‘states’ in multiple dimensions, which may in turn recruit distinct partners for computing beliefs regarding their identity.’

-Would it be helpful to show block-start tone related responses of DA? I suspect there may be no difference in DA activation between start of small reward block and large reward block (though they alternate and mice should be able to predict which block he will be in?). Is that true? Have the authors considered the meanings of this type of finding (or other types of block-start effects)? A previous paper (Bromberg-Martin, 2010, Neuron) analyzed trial start activity and was able to show a more complex anticipatory role of DA neurons as animals (macaques) expected belief and value state changes (from long reward to long punishment states/blocks). Of note, those data seem highly relevant here.

Response: We thank the Reviewer for this suggestion. We analyzed anticipatory licking and dopamine activity on sound presentation (Figure R9). Both showed some predictive change in block contingency:

- 1) Mice tended to increase licking after the sound came on following a small block;
- 2) At sound presentation, dopamine activity increased slightly following a small block and decreased following a big block.

This activity on sound presentation (i.e. block start) suggests that mice expected a switch in contingency, however, when the odor came on for the first trial, they licked and their dopamine neurons fired as if that switch was not (or less) expected.

This intricate pattern is indeed reminiscent of Ethan Bromberg-Martin and collaborators work in macaque monkeys, and is in line with what the Reviewer suggests as a more complex anticipatory role of dopamine neurons as animals expected belief and value state changes.

Figure R8. Anticipatory licking and dopamine activity at block transitions. (a, c) Anticipatory licking and GCaMP activity at block transitions. Data is separated based on the previous block (s_1 and s_2). The sound cue signals block start and is followed by the odor cue for trial 1. (b, d) Anticipatory licking and GCaMP activity are quantified over 1 second bins. Data represents mean \pm s.e.m. * $p > 0.05$ for post-hoc paired Wilcoxon tests. $n = 11$ mice

We have added figure R8 as Supplementary Figure 3, and have made the following changes to the manuscript to refer to this block start activity:

Results section, Page 6, Lines 96-106:

‘Analysing the licking and dopamine activity at block start, when the sound comes on, mice appeared to increase licking following the small block s_1 between sound offset and trial 1’s odor onset (during a fixed period of 3 seconds) (Supplementary Fig. 3 a, b). Although this was not sufficient to actually reverse the licking pattern on trial start, it likely contributed to the observed change in licking between trial 5 and 1 (Fig. 2b). Dopamine activity showed the opposite tendency, with decreasing activity following blocks s_2 (Supplementary Fig. 3 c, d). This activity on block start indicated that mice partially predicted a change in contingency, following the task’s initial training structure (deterministic switch between blocks during the first 10 days). However this predictive activity did not override the effect of the previous block on dopamine activity on cue

presentation as it was most similar to the activity on the preceding block's last trial (Fig. 2e).'

Discussion section, Page 12, Lines 237-241:

'Interestingly, both anticipatory licking and dopamine activity appeared to predict a switch in contingency upon block start. Although the amplitude of these pre-emptive changes were relatively small compared to responses to the odor cue and reward presentations, it indicated that the task structure influenced both behavior and dopamine activity, as had been previously shown in macaques²⁹.'

-I don't have experience with fiber photometry. So, one question that may be easy to answer but that arose in my reading of your paper (and that I have no online experience with) is whether the baseline of DA activity changes as the trials continue in a given block? This may be in Supplemental, or elsewhere, but I missed it. I think that would be worth reporting and on a very related note, more details about how you "normalize" (e.g. min-max) would be helpful as well.

Response: Thank you for pointing out the issue of baseline. Measuring baseline activity is not very easy with photometry due to technical limitations such as bleaching of calcium indicators. Therefore, it is difficult to make strong conclusions about the baseline firing across trials, and we prefer not to make inferences based on baseline signals in the present study. On the other hand, it is important to show that our main results are not affected by our estimates of baseline. To address this issue, we have tested 3 methods of baseline corrections:

1) In the current manuscript (reproduced in Figure R9a), GCaMP activity is presented as the relative change in fluorescence, $dF/F = (F-F_0)/F_0$, where the baseline F_0 is the mean fluorescence during a 1 s period before the odor presentation. With this correction, the fluorescence measured at each time point within a trial is corrected by the average fluorescence during a 1 sec period before odor presentation for that given trial. This is the activity illustrated in Figure 2d (notice how dF/F is at 0% over the first second before odor onset).

2) We additionally tested baseline correction in which F_0 is obtained from the 1 second period before block start, i.e. before odor onset (Figure R9b). With this correction, the fluorescence measured at each time point within a trial is corrected by the average fluorescence during a 1 sec period before sound presentation for that given block (i.e. over 5 consecutive trials).

3) Finally, we tested a baseline correction in which F_0 is defined as the median over a 60 sec window (Figure R9c). With this correction, the fluorescence measured at each time point is corrected by the median fluorescence over a 60 sec period centred around the given time point.

Two further normalizations were performed on the data, regardless of the baseline correction used:

1) First, when analyzing the reward (US) response, since the CS response did not always go back to baseline before reward presentation, US responses were baseline-corrected by subtracting the mean dF/F over the 100 ms period centered around US onset. This provided the measure for the actual change in activity at reward presentation.

2) Second, because the amplitude of GCaMP signals varied across the different recording conditions (mostly between mice expressing GCaMP under transgenic vs viral construct, see Figure R1, upper panels), the data were further corrected through min-max normalization for illustration purposes when plotting the summary data. We chose to normalize the signals from each individual mouse using trial 1's response as reference for the minimum and maximum values for the min-max normalization ($y = (x - \min_{\text{trial1}}) / (\max_{\text{trial1}} - \min_{\text{trial1}})$). This normalization corrects for the differences in amplitude across mice, while preserving the features observed in each recording condition, i.e. monotonicity in trial 1 and non-monotonicity in trial 2 (see Figure R1, lower panels).

Figure R9. Dopamine activity using different baseline correction methods (a) Trial level baseline correction, using 1 second before odor onset as baseline. (b) Block level baseline correction, using 1 second before sound onset as baseline. (c) Running median baseline correction, using the median over a 60 second period centred on the current data point analysed as baseline. Data represents mean \pm s.e.m. $n = 11$ mice

We observe the same patterns across baseline correction methods, supporting the fact that overall, our main conclusions are robust to distinct baseline correction methods, as well as normalization methods.

Although we believe that photometry is limited in its ability to estimate baseline activity, we used the running median obtained from the third baseline correction method as a proxy to address whether there was a change in baseline across block types. Within each recording session, we analysed the difference between the mean median in small and big blocks (Figure R10). Over mice, we did not see a significant difference between the blocks median fluorescence activity, suggesting that baseline activity did not significantly differ across blocks.

Figure R10. Analysis of median fluorescence levels across small (s_1) and big (s_2) blocks. Median raw fluorescence over 60 sec windows within small blocks were subtracted from medians over 60 sec windows within big blocks. No significant differences were observed across mice (t-test, $p = 0.24$). Data represents mean \pm s.e.m. $n = 11$ mice

We now discuss the issues related to baseline analysis limitation in Methods and clarified the normalization applied. We also added Figure R9 as Supplementary Figure 8.

Methods section, Page 18, Line 366-395 :

‘For GCaMP activity, we focused our analysis on the phasic responses. Indeed, a majority of previous work has shown reward prediction error-like signaling in the phasic responses of dopamine neurons and technical limitations such as bleaching limit our ability to monitor long-timescale changes in baseline using calcium imaging. Fluorescence data was acquired at 1 kHz. For each trial, the relative change in fluorescence, $dF/F = (F-F_0)/F_0$, was calculated by taking F_0 to be the mean fluorescence during a 1 s period before the odor presentation, such that the fluorescence measured at each time point within a trial is corrected by the average fluorescence during the 1 sec period before odor presentation for that given trial. We further tested two additional baseline normalizations to verify that our conclusions were robust with regards to the baseline normalization method (Supplementary Figure 8): (1) using as F_0 the 1 second period before block start, i.e. before sound onset, such that the fluorescence measured at each time point within a trial is corrected by the average fluorescence during the 1 sec period before sound presentation for that given block (i.e. over 5

consecutive trials); (2) using as F0 the median over a 60 sec window, such that the fluorescence measured at each time point is corrected by the median fluorescence over a 60 sec period centred around that given time point.

Mean GCaMP activity during odor (CS) and reward (US) presentations was calculated for each trial as the mean activity during the 1 s period after event onset. Two further normalizations were performed on the data, regardless of the baseline correction used:

1) When analysing the reward (US) response, since the CS response did not always go back to baseline before reward presentation, US responses were baseline-corrected by subtracting the mean dF/F over the 100 ms period centred around US onset. This provided a measure for the actual change in activity at reward presentation.

2) Since the absolute level of fluorescence was variable across mice that expressed GCaMP6f through viral injection or transgenetically (Supplementary Figure 4), for illustration purposes to summarize the data in one plot, each mouse's mean US response across rewards was normalized by min-max normalization when pooled together. The normalization was performed within each mouse, using the given mouse's trial 1 response as reference for the minimum and maximum values for the min-max normalization such that $y = (x - \min_{\text{trial1}}) / (\max_{\text{trial1}} - \min_{\text{trial1}})$ (Fig. 2 b and e, Supplementary Fig. 5, Supplementary Fig. 6, Supplementary Fig. 7). Of note, the models were not fit on the normalized data.'

Results section, Page 8, Lines 152-154:

'Lastly, the pattern of dopamine responses was observed independently of the baseline correction method we used, whether it was pre-trial, pre-block or using a running median as baseline (Supplementary Fig. 8).'

Minor concerns addressing style and clarity

-Figure 1 - the authors state "then intermediate rewards are introduced". This is a totally vague description of the crucial manipulation and does not give the clever design any credit. Please elaborate and describe the task in great detail in the legend (within the word limits of Nat. Com).

Response: We thank the Reviewer for the kind remark regarding the design. We have corrected the legend:

'Figure 1. Task design to test the modulation of dopaminergic RPEs by state inference. a Mice are trained on two perceptually similar states only distinguished by their rewards: small (s_1) or big (s_2). The different trial types, each starting by the onset of a unique odor (conditioned stimulus, CS) predicting the delivery of sucrose (unconditioned

*stimulus, US), were presented in randomly alternating blocks of five identical trials. A tone indicated block start. Only one odor and one sound cue were used for all blocks, making the two states perceptually similar prior to reward delivery. To test for state inference influence on dopaminergic neuron signalling, we then introduced rare blocks with intermediate-sized rewards. **b** RPE across varying rewards computed using standard reinforcement learning (RL). Because the same odor preceded both reward sizes, a standard RL model with a single state would produce RPEs that increase linearly with reward magnitude. **c** Belief state b across varying rewards defined as the probability of being in $s1$ given the received reward. **d** RPE across varying rewards computed using the value of the belief state b . A non-monotonic pattern across increasing rewards is predicted when computing the prediction error on the belief state b .'*

-Extended data Figures 2-3 are very important figures. I would suggest moving many of the panels from them to the main body of the manuscript.

Response: Regarding Supplementary Figure 2, as mentioned above, we appreciate the interest for our control data, which was obtained in one mouse, replicating previous published data from our lab using electrophysiological recording from optogenetically-identified dopamine neurons (Tian et al., 2015). Although fiber photometry data further extend these results, we feel that this is out of the scope of the present study and not sufficiently supported in terms of subject number to be in the main body of the manuscript.

On the other hand, we agree that former Supplementary Figure 3 (the training data) can be part of the main manuscript and is now inserted in Figure 2.

-Sentence on line 34 needs to be elaborated for clarity. It is a key point to the authors.

-Line 55 of introduction - I would front load this so that the rest is easy and exciting to read. It may increase the "word count" but in Nature Communications this is less of an issue.

Response: Thank you for these two suggestions. We have modified the introduction accordingly:

Introduction section, Page 3, Line 30-42 :

'Normative theories propose that animals represent their state uncertainty as a probability distribution or "belief state"¹⁻⁴ providing a probabilistic estimate of the true state of the environment based on the current sensory information. Specifically, optimal

state inference as stipulated by Bayes' rule computes a probability distribution over states (the belief state) conditional on the available sensory information. Such probabilistic beliefs about the current's state identity can be used to compute reward predictions by averaging the state-specific reward predictions weighted by the corresponding probabilities. Similarly to the way reinforcement learning (RL) algorithms update values of observable states using reward prediction errors, state-specific predictions of ambiguous states can also be updated by distributing the prediction error across states in proportion to their probability. This leads to the hypothesis that dopamine activity should reflect prediction errors computed on belief states. However, direct evidence for this hypothesis remains elusive. Here we examined how dopamine RPEs and subsequent learning are regulated under state uncertainty.'

Reviewers' comments:

Reviewer #1 (Remarks to the Author):

The authors tried to address my concerns through data reanalysis and clarification. Now their model matches better with the experimental data using the new normalization method. By clarification, the authors also resolved some of my prior minor concerns (e.g., minor concerns 2 & 3). However, most of my prior main concerns have not been satisfactorily addressed.

I had noted that the authors pooled data from 11 mice with different GCaMP expression levels (transgenic mice or viral expression) and recording sites (the VTA or the striatum). The response amplitudes were very variable between soma recordings and terminal recordings and even within the two terminal recording experiments. Moreover, the authors did not carry out necessary GFP control. Because the entire argument was based on this thin set of experimental data, I had asked the authors to "perform the experiments more rigorously". The authors responded by stating that the experiments would take too long, although it is clear that all experiments had been feasible from their initial submission and new recordings would have been done within the revision period.

Similarly, in the revision the authors used data from other studies to examine the relation between Ca²⁺ signals and neuronal spiking. They concluded that these data "support relatively linear relationship between spikes and photometry signals". The data quality shown in Figure R2a and R2b (others data) are much higher than what the authors presented in their own study. Therefore, their argument is not very convincing unless they provide data of similar quality. Overall, I feel very uncomfortable to do deep data analysis and model fitting without proper quality control of the experimental data.

Minor concerns.

(1) I am pleased to see that the model fits better with the re-normalized data for all trials (from 1 to 5). However, I am quite confused by the statement "Of note, the models were not fit on the normalized data."

(2) The authors "chose to quantify GCaMP responses as the average activity over 1 second following reward presentation (equivalent to the area under the curve)". They focused their "analysis on the phasic responses", although it remains unclear whether the average activity accurately reports the phasic response. The peak GCaMP6 response amplitude may reflect the phasic response more accurately than the average activity. I recommend the authors to reanalyze their data using peak responses, investigate whether the model fits or not, and compare the difference between the two different analysis methods.

Reviewer #2 (Remarks to the Author):

The authors did a fantastic job responding to my comments. I now believe this paper should be published in Nature Communications.

I do believe that with time and appropriate studies, such as this, we will come to an important "point" in research, at which we will have to speak to the following issue: many different neuromodulatory neurons signal "RPE-like" signals, have common projections, and also signal other parameters related to beliefs and sensory events (such as rewards, which also modulate value estimates). What really are differences and similarities between modulatory systems? What do they do at the synaptic level? etc. I think this paper will sit nicely in this evolution and provide important information.

Point-by-point response

In the following, the reviewers' comments are in slab-serif font (*Courier New*), our response appears in sans-serif font (*Arial*) and text from the revised manuscript and previous publications appear in *italic*.

Reviewer #1 (Remarks to the Author):

The authors tried to address my concerns through data reanalysis and clarification. Now their model matches better with the experimental data using the new normalization method. By clarification, the authors also resolved some of my prior minor concerns (e.g., minor concerns 2 & 3).

Response: We thank the Reviewer for these comments.

However, most of my prior main concerns have not been satisfactorily addressed.

I had noted that the authors pooled data from 11 mice with different GCaMP expression levels (transgenic mice or viral expression) and recording sites (the VTA or the striatum). The response amplitudes were very variable between soma recordings and terminal recordings and even within the two terminal recording experiments. Moreover, the authors did not carry out necessary GFP control. Because the entire argument was based on this thin set of experimental data, I had asked the authors to "perform the experiments more rigorously". The authors responded by stating that the experiments would take too long, although it is clear that all experiments had been feasible from their initial submission and new recordings would have been done within the revision period.

Response: The reviewer here brings up two points: (1) pooling data from mice with different recording strategies and (2) GFP controls.

Regarding point (1), we wish to clarify that our main conclusions are based on two data sets ($n = 5$ and $n = 4$ mice). We additionally obtained another data set from 2 animals. The imaging methods differed slightly between these three data sets, yet **similar results were obtained in each condition** except for fully expected differences in signal intensity (Supplementary Figure 5, e.g. transgenic vs virus expression levels). Each animal performed many trials (1650 ± 235 trials) over several experimental sessions, and 170 ± 25 trials of probe trials were obtained from each animal. In this type of experiment, given that we are measuring calcium signals in many trials per animal, sample sizes of 4 and 5 animals are not small. Importantly, these three recording conditions exhibited the **same response patterns**, both at the **group** level (see

Supplementary Figure 5, reproduced below) and, most importantly, at the **individual** level (see Supplementary Figure 7d, reproduced below).

Note that in all cases we used the **same calcium reporter**, GCaMP6f (Chen et al. 2013). Regarding the difference in expression strategy, the direct consequence is a difference in signal intensity: transgenic expression levels are lower than with viral vectors. Regarding the difference in recording site (cell body vs terminals), we have previously published that distinct dopaminergic neurons encode prediction errors differently based on their projection site (Menegas et al, 2017). In this paper, our aim was to record from the population, which signals the canonical reward prediction error, i.e. the medial VTA population projecting to the ventral striatum. We here targeted the same or highly overlapping population, either recording the cell bodies or the terminals. It was thus expected that we would observe the same pattern. By performing the recordings at these two sites, we were able to verify this hypothesis, as well as reject the hypothesis about potential differential modulation of activity at cell bodies versus terminals in our task.

We recognize that, to the best of our knowledge, no study has performed recordings in different settings with the final aim of pooling the data, since this strategy is usually pursued in studies aiming at unravelling functional differences (e.g. Lerner et al. 2015, Parker et al. 2016, Menegas et al. 2017). In our case we interpret this as a sign of robustness of the modulation of canonical dopaminergic prediction error by belief states.

Importantly, our model fitting, which allows us to test the hypothesis of belief state modulation of the dopaminergic activity, was performed on each mouse individually (point further addressed in Reviewer 1's first minor comment). Hence no pooling across recording conditions was involved in the model fitting. Although we do not believe the pooling of the data would affect the model fits given that the non-monotonic pattern is observed across all mice, this concern here is unwarranted since models were fit individually.

We thus have in total 11 animals, which exhibit the same pattern, whether we record from viral or transgenic expression or cell bodies or terminals. 11 is more than a standard sample size in similar experiments (Lerner et al. 2015, Parker et al. 2016, Matias et al., 2017) and we believe the existing data sufficiently supports our conclusions. Please note also that all of our main conclusions are supported by statistical analyses.

Supplementary Figure 5. Dopamine responses on trials 1 and 2 plotted separately based on recording conditions. *a, d* Dopamine responses in mice expressing transgenetically GCaMP6f in DAT-positive neurons and recorded from VTA cell bodies ($n = 5$). *b, e* Dopamine responses in mice expressing GCaMP6f through a viral construct in DAT-positive neurons and recorded from VTA cell bodies ($n = 2$). *c, f* Dopamine responses in mice expressing GCaMP6f through a viral construct in DAT-positive neurons and recorded from dopamine neuron terminals in the ventral striatum ($n = 4$). The upper row (*a - c*) shows the average across mice, while the lower row (*d - f*) shows the same average after normalizing within mice through min-max normalization using trial 1's response as reference for the minimum and maximum values. This normalization corrects for the different amplitudes in GCaMP signals across the different recording conditions, but preserves the features observed in each recording condition. Note that the monotonicity and non-monotonicity of the responses in trials 1 and 2, respectively, are observed in each recording condition (*a - c*). Data represents mean \pm s.e.m.

Supplementary Figure 7. d Examples of individual dopamine responses across trials. Data represents mean \pm s.e.m.

(2) With respect to GFP controls, we have previously reported that motion artefacts are almost none and negligible in our experimental conditions using

head-fixed preparations in mice (Menegas et al., 2017). Indeed using a set-up with 473 nm and 561 nm lasers to deliver light to excite respectively GFP and tdTomato reporters, we observed large responses to unpredicted reward in GCaMP, but not tdTomato, signals (Menegas et al., 2017). Because the present study uses essentially the same head-fixation, task, and imaging system, we do not expect that this is an issue. Furthermore, it is not trivial to obtain the non-monotonic response functions observed in the present study. Indeed, the main cause of motion artefact in head-fixed settings, licking, tends to monotonically increase as a function of presented reward sizes, rather than exhibiting a non-monotonic (s-shaped) modulation (see Figure R1).

Figure R1. Dopamine responses (top) and licking (bottom) at reward delivery on trial 2. Dopamine responses and licking were quantified identically, as the average activity during 1 second following reward delivery. Note that for bigger intermediate rewards dopamine activity decreases, unlike licking activity. (a, d) Mice expressing transgenetically GCaMP6f in DAT-positive neurons and recorded from VTA cell bodies (n = 5). (b, e) Mice expressing GCaMP6f through a viral construct in DAT-positive neurons and recorded from VTA cell bodies (n = 2). (c, f) Mice expressing GCaMP6f through a viral construct in DAT-positive neurons and recorded from dopamine neuron terminals in the ventral striatum (n = 4). Data represents mean \pm s.e.m.

Similarly, in the revision the authors used data from other studies to examine the relation between Ca²⁺ signals and neuronal spiking. They concluded that these data "support relatively linear relationship between spikes and photometry signals". The data quality shown in Figure R2a and R2b (others data) are much higher than what the authors presented in their own study. Therefore, their argument is not very convincing unless they provide data of similar quality. Overall, I feel very

uncomfortable to do deep data analysis and model fitting without proper quality control of the experimental data.

Response: The linearity of photometry signals is justified using our unpublished data in the rebuttal (Figure R3). Importantly, even in the low dF/F regime, firing rate and photometry signals show a similar trend. This data is from a different manuscript, which is currently under review in a different journal. We therefore cannot use this data set in the current manuscript.

[Redacted]

Yet Reviewer 1 states that ‘the data quality shown in Figure R2a and R2b (others data) are much higher than what the authors presented in their own study’. However, this claim is not justified. The smaller signal intensity in the present study was mainly because the data shown in the rebuttal figures is from unexpected reward responses. In the present study, we plotted the reward responses when the reward was expected. Therefore, the magnitudes of reward responses are expected to be small compared to unexpected reward – the very characteristic of reward prediction error responses.

Nonetheless, addressing the Reviewer’s concern about data quality, we can compare responses obtained in mice from this current study to the data presented in the Figure R3, comparing the activity in similar conditions, i.e. ventral striatum dopamine neuron terminals with viral expression responses to unexpected delivery of similar sized reward. Note that the absolute intensity of signals varies by various factors not related to signal-to-noise ratio. To emphasize the reliability of signals, we here plotted the data from the two studies using z-scores. We observe a similar pattern (Figure R4), indicating the similar signal reliabilities across the studies. Therefore, there is no reason to believe that the data in the current study is noisier or less reliable.

Figure R4. Recordings of ventral striatum dopamine neuron terminals to unexpected reward delivery. Traces from (a) Menegas et al., submitted, replotted from figure R2 and (b) from the 4 mice with VS terminal recordings in Babayan et al. The mice are trained in different tasks between Menegas et al. and Babayan et al. and recorded in different rigs, yet the response profiles are similar in these similar trial types. Data represents mean \pm s.e.m.

Minor concerns.

(1) I am pleased to see that the model fits better with the re-normalized data for all trials (from 1 to 5).

However, I am quite confused by the statement "Of note, the models were not fit on the normalized data."

Response: Reviewer 1 refers to the Methods section, where we detailed the min-max normalization we employed (copied below). As mentioned in the methods section, this min-max normalization's sole use is for illustration purposes to summarize the data across mice in one plot. Following Reviewer 1's initial suggestion, we indeed modified this normalization by using trial 1's response as reference for the minimum and maximum values (i.e. $y = (x - \min_{\text{trial1}}) / (\max_{\text{trial1}} - \min_{\text{trial1}})$), instead of our originally misguided normalization performed independently for each trial, which prevented us from visualizing amplitude changes across trials. Yet, since the models are fit on each mouse individually, there is no need to min-max normalize the data since the GCaMP activity within mice is constant. We pointed out at the end of the Methods section on normalization that "the models were not fit on the normalized data." to clearly specify what data was used for the model fitting. We wish to keep this sentence but acknowledge it may have been confusing. The following correction will hopefully make this point clearer: '*Of note, the models were not fit on the min-max normalized data but directly on mice's individual baseline-corrected GCaMP activity.*'

Since the absolute level of fluorescence was variable across mice that expressed GCaMP6f through viral injection or transgenetically (Supplementary Figure 4), for illustration purposes to summarize the data in one plot, each mouse's mean US response across rewards was normalized by min-max normalization when pooled together. The normalization was performed within each mouse, using the given mouse's trial 1 response as reference for the minimum and maximum values for the min-max normalization such that $y = (x - \min_{\text{trial1}}) / (\max_{\text{trial1}} - \min_{\text{trial1}})$ (Fig. 2 b and e, Supplementary Fig. 5, Supplementary Fig. 6, Supplementary Fig. 7). Of note, the models were not fit on the min-max normalized data but directly on mice's individual baseline-corrected GCaMP activity.'

(2) The authors "chose to quantify GCaMP responses as the average activity over 1 second following reward presentation (equivalent to the area under the curve)". They focused their "analysis on the phasic responses", although it remains unclear whether the average activity accurately reports the phasic response. The peak GCaMP6 response amplitude may reflect the phasic response more accurately than the average activity. I recommend the authors to reanalyze their data using peak responses, investigate whether the model fits or not, and compare the difference between the two different analysis methods.

Response: We reproduced our data analysis and model fitting to the peak GCaMP response and verified that the results obtained were similar (Figure R5 and Table R1). We now include this analysis in our manuscript as Supplementary Figure 6 and Supplementary Table 2, and modified the results sections to include references to this analysis.

'Similar results were obtained when measuring the peak response following reward presentation instead of the average activity over 1 second (Supplementary Fig. 6, a - g).'

'Bayesian information criterion (BIC) and random-effects model selection^{21,22} computed on each of the six models fit to individual mice's dopamine activity both favoured the RL model with belief states with two initial free priors over other models, in particular over the standard RL model with two free initial values (Supplementary Table 1; Supplementary Fig. 8c). Similar results were obtained when fitting the peak GCaMP response after reward presentation (Supplementary Table 2; Supplementary Fig. 6h).'

Figure R5 . Data and model fits on peak dopamine response. Quantifying the peak dF/F response following reward delivery recapitulates the results obtained by quantifying the average response over one second post reward delivery. **(a, d)** Dopamine responses on trials 1 and 2 for mice expressing transgenetically GCaMP6f in DAT-positive neurons and recorded from VTA cell bodies (n = 5). **(b, e)** Dopamine responses on trials 1 and 2 for mice expressing GCaMP6f through a viral construct in DAT-positive neurons and recorded from VTA cell bodies (n = 2). **(c, f)** Dopamine responses on trials 1 and 2 for mice expressing GCaMP6f through a viral construct in DAT-positive neurons and recorded from dopamine neuron terminals in the ventral striatum (n = 4). The upper row **(a - c)** shows the average across mice, while the lower row **(d - f)** shows the same average after normalizing each mouse's signal by min-max normalization. This normalization corrects for the different amplitudes in GCaMP signals across the different recording conditions, but preserves the features observed in each recording condition. Note that the monotonicity and non-monotonicity of the responses in trials 1 and 2, respectively, are observed in each recording condition **(a - c)**. **(g)** Normalized dopamine responses for all mice on trials 1 to 5. **(h)** Best fit by standard and with belief state reinforcement learning models. Data represents mean \pm s.e.m.

Model		Number of parameters	Parameters	Parameter estimates	Log-likelihood	BIC	Exceedance probability	Protected exceedance probability
Standard reinforcement learning	1 state, 1 fixed initial value (0.5)	1	learning rate (α)	0.257	-57.4472	118.989	0.0379	0.1025
	1 state, 2 initial values depending on previous block (model in Fig. 3)	3	learning rate (α)	0.2915	-50.6742	113.631	0.3238	0.245
			value following s_1 (V_1)	0.0063				
value following s_2 (V_2)	0.3383							
Reinforcement learning with belief state	2 states, 1 fixed initial prior (0.5)	2	learning rate (α)	0.0282	-59.8138	127.816	0.0064	0.0868
			sensory noise variance (σ)	0.4597				
	2 states, 1 initial prior	3	learning rate (α)	0.2711	-52.9573	118.197	0.0113	0.0893
			sensory noise variance (σ)	0.239				
			prior following s_1 (p_1) (with prior following s_2 $p_2=1-p_1$)	0.948				
	2 states, 2 initial priors depending on previous block (model in Fig. 3)	4	learning rate (α)	0.27	-46.5757	109.529	0.5774	0.3713
			sensory noise variance (σ)	0.315				
			prior following s_1 (p_1)	0.973				
			prior following s_2 (p_2)	0.556				
	3 states, 2 initial priors depending on the previous block and 1 for the intermediate states	5	learning rate (α)	0.046	-48.2943	117.06	0.0432	0.1051
sensory noise variance (σ)			0.156					
prior following s_1 (p_1)			0.934					
prior following s_2 (p_2)			0.0026					
prior for intermediate rewards (p_r)	0.389							

Table R1. Best-fitting parameter estimates shown as mean across mice and model comparison on peak GCaMP response. Bayesian information criterion (BIC) and exceedance probabilities (Stephan et al. 2009; Rigoux et al., 2014) both favoured the RL model with belief states with two initial free priors over other models. The best values are highlighted in bold.

Reviewer #2 (Remarks to the Author):

The authors did a fantastic job responding to my comments. I now believe this paper should be published in Nature Communications.

I do believe that with time and appropriate studies, such as this, we will come to an important "point" in research, at which we will have to speak to the following issue: many different neuromodulatory neurons signal "RPE-like" signals, have common projections, and also signal other parameters related to beliefs and sensory events (such as rewards, which also modulate value estimates). What really are differences and similarities between modulatory systems? What do they do at the synaptic level? etc.

I think this paper will sit nicely in this evolution and provide important information.

We thank the Reviewer for their comment and appreciation of the contribution of our work to the field.

Reviewers' comments:

Reviewer #4 (Remarks to the Author):

This paper measures how RPE represented by DA neurons varies as a function of reward size, in a paradigm where the mouse is trained to expect some rewards (1 or 10 uL), but not others (values in between 1 and 10uL). The general aim is then to infer from the shape of this function whether the DA signal reflect "prediction error computed on belief states.

I found this general aim conceptually interesting, though somewhat semantic (is it really important to distinguish between "prediction" and "belief" ? These terms seem synonymous. No strong case is made in the paper why it is important to make this distinction, either for the animal or for artificial RL agents).

The techniques used were also appropriate for addressing this aim.

Major:

The author's main argument seems to be largely based on this assumption:

"Because the same odor preceded both reward sizes, a standard RL model with a single state would produce RPEs that increase linearly with reward magnitude"

I do not understand how this assumption (illustrated in fig. 1b) is justified:

It seems that the authors train the mice to expect, in response to odor cue, EITHER 1 uL OR 10 uL. The classic RL theory would then predict that the RPE for either of these rewards should be low. Then the intermediate-sized rewards should produce an RPE that is mostly higher than the latter (1 or 10 uL) RPE. This is largely what the authors find in Fig. 3g. So the results are not substantially different from simple predictions of classic RL theory, and thus it is not clear if the authors found anything new.

Unless the authors better explain/justify fig. 1b, i.e. why they expect one expected reward (1uL) to have a lower RPE than another equally expected reward (10 uL), the logic of the paper makes no sense to me.

Minor:

The experimental data do not really resemble the authors' prediction in fig. 1d: the DA signal in Fig. 3 never dips below 0 (but 1d does). So the data do not really match the prediction, except in the trivial ("non-monotonic") sense. This should be better discussed.

Point-by-point response

We would like to thank Reviewer #4 for her/his comments, which we address below by point-by-point response as well as in the accompanying manuscript. In the following, the reviewer's comments are in slab-serif font (*Courier New*), our response appears in sans-serif font (Arial) and text from the revised manuscript appear in *italic*.

Reviewer #4 (Remarks to the Author)

This paper measures how RPE represented by DA neurons varies as a function of reward size, in a paradigm where the mouse is trained to expect some rewards (1 or 10 uL), but not others (values in between 1 and 10uL). The general aim is then to infer from the shape of this function whether the DA signal reflect "prediction error computed on belief states.

I found this general aim conceptually interesting, though somewhat semantic (is it really important to distinguish between "prediction" and "belief"? These terms seem synonymous. No strong case is made in the paper why it is important to make this distinction, either for the animal or for artificial RL agents).

Response:

Thank you for pointing this out. We here wish to clarify what we mean by 'prediction' and 'belief'.

(1) We use the terms, 'reward prediction' and 'belief state' as commonly used in the reinforcement learning field. The former refers to estimating a reward (in our case, reward size specifically), which can be done from any given state. The latter refers to inferring the likelihood of being in a certain state based on ambiguous stimuli.

The difference between belief and prediction is thus not semantic. They formally refer to two distinct non-overlapping processes: learning an accurate *state representation* (in this case either a standard RL sensory-defined state or a Bayesian belief state), then using the state representation to *predict* rewards.

(2) As the reviewer indicates, in our paradigm, because reward amount defines states, reward prediction and belief state are correlated. However, we can still make meaningful theoretical distinctions. Indeed we explored and tested several different models, which made qualitatively different predictions, and we provide clear support for one of them only.

(3) Importantly, the main point of our work is not to distinguish neural correlates of prediction versus belief. As further detailed in our response to the major point, the main distinction between conventional and novel classes of models is the following: the conventional model does not have distinct states corresponding to the small and large reward states, and reward prediction is based on the cached value learned directly from experienced reward, whereas the belief state model has distinct states corresponding to the small and large reward states. In the latter case, the animal or agent uses ambiguous information to infer which state it is in, and predicts reward based on this inferred state (the belief state).

To clarify these points, we have revised the manuscript.

Introduction section, page 3, line 26-44:

'Dopamine neurons are thought to report a reward prediction error (RPE, or the discrepancy between observed and predicted reward) that drives updating of predictions¹⁻⁵. In reinforcement learning (RL) theories, future reward is predicted based on the current state of the environment⁶. Although many studies have assumed that the animal has a perfect knowledge about the current state, in many situations the information needed to determine what state the animal occupies is not directly available. [...] Similarly to the way RL algorithms update values of observable states using reward prediction errors, state-specific predictions of ambiguous states can also be updated by distributing the prediction error across states in proportion to their probability. Simply put, standard RL algorithms compute reward prediction on observable states, but under state uncertainty reward predictions should normatively be computed on belief states, which correspond to the probability of being in a given state. This leads to the hypothesis that dopamine activity should reflect prediction errors computed on belief states.'

Results section, page 5, line 84-92:

In our paradigm, because reward amount defines states, reward prediction and belief state are closely related. Yet with the same reward amount, standard RL and belief state RL make qualitatively different predictions (Fig. 1b vs Fig. 1d). The main distinction between both classes of models is the following: the standard RL model does not have distinct states corresponding to the small and large reward states, and reward prediction is based on the cached value learned directly from experienced reward, whereas the belief state model has distinct states corresponding to the small and large reward states (Supplementary Fig. 1, left column). In the latter

case, the animal or agent uses ambiguous information to infer which state it is in, and predicts reward based on this inferred state (i.e. belief state).

The techniques used were also appropriate for addressing this aim.

Major:

The author's main argument seems to be largely based on this assumption: "Because the same odor preceded both reward sizes, a standard RL model with a single state would produce RPEs that increase linearly with reward magnitude"

I do not understand how this assumption (illustrated in fig. 1b) is justified: It seems that the authors train the mice to expect, in response to odor cue, EITHER 1 uL OR 10 uL. The classic RL theory would then predict that the RPE for either of these rewards should be low. Then the intermediate-sized rewards should produce an RPE that is mostly higher than the latter (1 or 10 uL) RPE. This is largely what the authors find in Fig. 3g. So the results are not substantially different from simple predictions of classic RL theory, and thus it is not clear if the authors found anything new. Unless the authors better explain/justify fig. 1b, i.e. why they expect one expected reward (1uL) to have a lower RPE than another equally expected reward (10 uL), the logic of the paper makes no sense to me.

Response:

What we are calling "standard RL" corresponds to the assumption that the state corresponds to the cue, which is the same for all trials. As noted by the reviewer, we trained the mice to expect either 1 μ L or 10 μ L, following an odor cue. Crucially, we only present one odor to the mice. Thus, at each trial start, the set of sensory inputs is identical, regardless of the amount of reward delivered. Under the 'cue = state' assumption (standard in the RL neuroscience literature), separate values cannot be learned for different reward amounts. Therefore, depending on the recent history, the value of the state will be either low or high. Importantly, this state value will be used as a reference point when mice are presented with an intermediate reward. The comparison between any intermediate reward and a unique value of the state will result in a monotonically increasing RPE. Keep in mind that these are signed prediction errors, so some will be negative and some will be positive, but the function will always be monotonic in reward. In other words, the actual

value of the state will affect the intercept of the linear RPE response, but not its monotonicity.

In Fig. 1b and Supp. Fig 1a, we illustrated our prediction with a state of average value 0.5 (on a scale between 0 and 1, this would be equivalent to 4.5 μL).

Since we noticed that mice pre-emptively anticipated a switch in reward size at block transition (Fig. 2b, a feature likely resulting from the early reversal training regime), we additionally considered a situation where the state may rapidly update its value at block start based on the previous block. In practice, we implemented this by allowing a different initial value for the state at block start depending on the previous block. This also illustrates the effect of the state's value on the intercept. Importantly, it also results in a monotonically increasing RPE, since, here again, all intermediate rewards are being compared to one value.

Now our critical theoretical development here is to introduce belief states. Since the sensory inputs are identical, and do not allow distinguishing the small reward from the big one upon odor presentation, a proposed strategy to solve the uncertainty inherently tied to the unique odor is by computing a belief about the current state's identity. Following the Bayesian framework, this belief is proportional to the product of the likelihood of the current sensory inputs ($P(r|s)$) by the prior about the likelihood of the occurrence of a given state ($P(s)$). This model explicitly assumed the existence of multiple states distinguished by their reward distributions, since the likelihood $P(r|s) = \mathbf{N}(r, \bar{r}_s, \sigma^2)$ was defined as a normal distribution over rewards r , centred on the average reward normally obtained in either the small or the big state (\bar{r}_s), with a sensory noise variance σ^2 . Thus, in spite of identical sensory inputs, prior experience allows probabilistically distinguishing several states (one associated to 1 μL and one to 10 μL). If mice or agents rely on a multi-state representation, they now have two reference points to compare the intermediate rewards to, leading to a non-monotonic reward prediction error pattern (Fig. 1d and Supplementary Fig. 1 c-e). Indeed, intermediate rewards closer to the small reward are more likely to be compared to it, whereas intermediate rewards closer to the big one are more likely to be compared to it.

Note that this non-monotonicity also holds if we allow for an additional belief state (Supplementary Fig. 1f).

RPE monotonicity is a consequence of a unique state representation, whereas RPE non-monotonicity results from a multi-state representation. Crucially, our use of a unique odor cue is what allows distinguishing

predictions from a standard RL theory from a belief-state based RL. Our results are thus different from simple predictions of standard RL theory.

Results section, page 4-5, line 65-77:

*To test for state inference influence on dopaminergic neuron signalling, we then introduced rare blocks with intermediate-sized rewards. Because the same odor preceded both reward sizes, a standard RL model with a single state (corresponding to the odor) would produce RPEs that increase linearly with reward magnitude (Fig. 1b, Supplementary Fig. 1a)^{10,11}. This prediction follows from the fact that the single state's value will reflect the average reward across blocks, and RPEs are equal to the observed reward relative to this average reward value. **The actual value of the state will affect the intercept of the linear RPE response, but not its monotonicity. In Fig. 1b and Supp. Fig 1a, we illustrated our prediction with a state s_t of average value 0.5 (on a scale between 0 and 1, which would be equivalent to 4.5 μ L).***

*A strikingly different pattern is predicted by an RL model that uses state inference to compute reward expectations. Optimal state inference is stipulated by Bayes' rule, which computes a probability distribution over states (the belief state) conditional on the available sensory information. **This model explicitly assumes the existence of multiple states distinguished by their reward distributions (see methods). Thus, in spite of identical sensory inputs, prior experience allows to probabilistically distinguish several states (one associated to 1 μ L and one to 10 μ L). If mice rely on a multi-state representation, they now have two reference points to compare the intermediate rewards to. Upon the introduction of new intermediate rewards, the probability of being in the state s_1 would be high for small water amounts and low for large water amounts (Fig. 1c). The subsequent reward expectation would then be a probability-weighted combination of the expectations for s_1 and s_2 . Consequently, smaller intermediate rewards would be better than the expected small reward (a positive prediction error) and bigger intermediate rewards would be worse than the expected big reward (a negative prediction error), resulting in a non-monotonic pattern of RPEs across intermediate rewards (Fig. 1d, Supplementary Fig. 1c).***

Minor:

The experimental data do not really resemble the authors' prediction in fig. 1d: the DA signal in Fig. 3 never dips below 0 (but 1d does). So the data do not really match

the prediction, except in the trivial (“non-monotonic”) sense. This should be better discussed.

Response:

Theoretically, a fully expected reward does not elicit a prediction error. Correspondingly, in 1997, Schultz and colleagues showed no change in dopamine neuron firing when their monkeys fully expected reward 1 second after the onset of the presentation of the predicting cue. Yet following studies in monkeys by Schultz and collaborators have repeatedly shown a response to expected reward when the reward was delivered over 2 seconds after cue onset (e.g. Fiorillo et al. 1998. Note that here we also have 2 seconds). Moreover, studies performed in mice in the Uchida lab and other labs (Geoffrey Schoenbaum, Ilana Wittten) also show this persistent positive response. Additionally to the effect of longer inter-stimulus interval (ISI), rodent studies using odors as timing onset cues do not allow a tight control between the onset of the sensory response to the cue, since it depends on when the first sniff occurs after odor onset. This small jitter in timing limits the full predictability of the reward delivery timing.

Because of these two factors (long ISI and sniff related timing jitter), we expected our dopamine reward responses to be generally shifted above 0 (note however that some responses did drop below 0 in some mice, see for example Supplementary Figure 8d, Mouse 4’s trial 2). This was accounted for in our modelling by fitting a coefficient β , which mapped theoretical prediction errors linearly to the measured dopamine response (i.e., the GCaMP signal) (Methods section line 457). Importantly, this does not change our prediction, which critically relies on the non-monotonicity of the dopamine response across increasing rewards.

Methods section, page 23, line 479-484:

All belief state models had a minimum of three free parameters: the learning rate α , the sensory noise variance σ^2 , and a coefficient β , which mapped theoretical prediction errors linearly to the measured dopamine response (i.e., the GCaMP signal). Indeed, because of a relatively long delay between odor onset and reward delivery (2 seconds), as well as timing jitter resulting from when mice first sniff after odor onset, we expected our dopamine reward responses to be generally shifted above 0^{3,4,55}. This was accounted for by fitting β